# Co-recruitment analysis of the CBL and CBLB signalosomes in primary T cells identifies CD5 as a key regulator of TCR-induced ubiquitylation

Guillaume Voisinne[1,†], Antonio García-Blesa[1,†], Karima Chaoui[2], Frédéric Fiore[3], Elise Bergot[1], Laura Girard[1,3], Marie Malissen[1,3], Odile Burlet-Schiltz[2], Anne Gonzalez de Peredo[2], Bernard Malissen[1,3,*] & Romain Roncagalli[1,**]

## Abstract

T-cell receptor (TCR) signaling is essential for the function of T cells and negatively regulated by the E3 ubiquitin–protein ligases CBL and CBLB. Here, we combined mouse genetics and affinity purification coupled to quantitative mass spectrometry to monitor the dynamics of the CBL and CBLB signaling complexes that assemble in normal T cells over 600 seconds of TCR stimulation. We identify most previously known CBL and CBLB interacting partners, as well as a majority of proteins that have not yet been implicated in those signaling complexes. We exploit correlations in protein association with CBL and CBLB as a function of time of TCR stimulation for predicting the occurrence of direct physical association between them. By combining co-recruitment analysis with biochemical analysis, we demonstrated that the CD5 transmembrane receptor constitutes a key scaffold for CBL- and CBLB-mediated ubiquitylation following TCR engagement. Our results offer an integrated view of the CBL and CBLB signaling complexes induced by TCR stimulation and provide a molecular basis for their negative regulatory function in normal T cells.

**Keywords** CBL; CBLB; CD5; ubiquitylation

**Subject Categories** Immunology; Post-translational Modifications, Proteolysis & Proteomics; Signal Transduction

**Mol Syst Biol. (2016) 12: 876**

## Introduction

Affinity purification followed by mass spectrometry (AP-MS) enables analysis of protein–protein interaction (PPI) networks (interactomes) and of signaling complexes (signalosomes) in their physiological cellular context (Gingras *et al*, 2007; Hein *et al*, 2015). Although such large-scale studies have resulted in comprehensive views of several signalosomes, they provided limited information on their internal organization and dynamics of assembly. Such information is, however, particularly valuable to understand how the flow of information resulting from engagement of a given cell surface receptor propagates over time (Collins *et al*, 2013; Zheng *et al*, 2013). The T-cell antigen receptors (TCRs) found at the surface of T cells carry out the formidable task of identifying minute number of antigenic peptides. Despite a wealth of information concerning the dozens of molecules involved in TCR signaling, the general principle of information processing by the TCR signal transduction network remains incompletely understood (Chakraborty & Weiss, 2014; Malissen & Bongrand, 2015).

TCR signals propagate through the cytosol and the nucleus via a multilayered signal transduction network involving PPI, phosphorylation, and ubiquitylation. Regulatory mechanisms limit and terminate TCR-driven signals allowing the immune system to return to a basal state after antigen has been cleared (Acuto *et al*, 2008). The E3 ubiquitin–protein ligases CBL and CBLB have been recognized as important negative regulators of T-cell activation (Huang & Gu, 2008). Following TCR engagement, the protein tyrosine kinases (PTK) LCK and ZAP-70 are activated and phosphorylate several transmembrane and cytosolic proteins, leading in turn to the recruitment of CBL and CBLB via their tyrosine kinase binding (TKB) domains or indirectly via the GRB2 adaptor. As a result, CBL and CBLB are rapidly phosphorylated on tyrosine residues, causing conformational changes allowing the recruitment of E2 ubiquitin-conjugating enzymes and their juxtaposition to substrates that include the TCR itself, scaffold proteins (also known as adaptors), cytosolic PTK, and phosphatases (Schmidt & Dikic, 2005). Modification of the TCR and of its signaling partners with ubiquitin promotes their sorting to multivesicular

1 Centre d'Immunologie de Marseille-Luminy, Aix Marseille Université, Inserm, CNRS, Marseille, France
2 Institut de Pharmacologie et de Biologie Structurale, Département Biologie Structural Biophysique, Protéomique Génopole Toulouse Midi Pyrénées, CNRS UMR 5089, Toulouse Cedex, France
3 Centre d'Immunophénomique, Aix Marseille Université UM2, Inserm US012, CNRS UMS3367, Marseille, France
*Corresponding author. Tel: +33 491269478; Fax: +33 491269430; E-mail: bernardm@ciml.univ-mrs.fr
**Corresponding author. Tel: +33 491269478; Fax: +33 491269430; E-mail: roncagalli@ciml.univ-mrs.fr
†These authors contributed equally to this work

bodies and lysosomal degradation, thereby ensuring termination of TCR signaling (Vardhana *et al*, 2010). CBL and CBLB are more than just E3 ubiquitin–protein ligases and possess numerous interactive elements that nucleate the formation of complexes thought to regulate cytoskeletal rearrangement (Lee & Tsygankov, 2013). These interactive elements comprise a proline-rich region that binds SH3 domain-containing proteins, multiple tyrosine phosphorylation sites enabling interactions with SH2 domain-containing proteins, and a ubiquitin association (UBA) domain.

Although the presence of several conserved structural domains in CBL and CBLB suggests that they exert redundant functions, the mature T cells of mice deprived of CBL or CBLB showed distinct phenotypes. Mature T cells from CBL-deficient mice are less responsive to antigenic stimulation (Naramura *et al*, 2002). In contrast, CBLB deficiency lowers the threshold that needs to be overcome by antigenic stimulation to trigger T-cell responses (Bachmaier *et al*, 2000; Chiang *et al*, 2000). As a result, CBLB-deficient T cells proliferate and synthesize cytokines following stimulation with suboptimal concentration of anti-TCR antibody or in the absence of CD28 costimulation, and in turn, CBLB-deficient mice are prone to autoimmunity (Gronski *et al*, 2004; Chiang *et al*, 2007; Teh *et al*, 2010; Zhou *et al*, 2014). CBLB-deficient T cells have increased anti-tumor efficacy, suggesting that CBLB might be exploited as a drug target for the purpose of boosting T-cell responses against tumors (Hinterleitner *et al*, 2012).

Here, we combined mouse genetics and time-resolved, quantitative proteomics analysis to compare the mode of action of CBL and CBLB following TCR-mediated activation of mature T cells. Considering that the transformed T-cell lines that have been used in previous AP-MS studies lacked several key signaling proteins (Astoul *et al*, 2001), we have developed mice that bear a genetic tag allowing AP-MS of the signalosomes that assemble around CBL and CBLB at different time points following physiological activation of primary CD4$^+$ T cells. 98 and 43 unique proteins were found to specifically interact with CBL and CBLB, respectively, in at least one condition of stimulation. Most of those interactions have not been described in the literature to our knowledge. To go beyond a mere inventory of the composition of the CBL and CBLB signalosomes and attempt to decipher the underlying PPIs responsible for their assembly, we further analyzed the kinetics of formation of such signalosomes. We showed that the correlations that exist among some preys in their kinetics of assembly with and disassembly from the CBL and CBLB baits over 600 s of TCR signaling were predictive of the probability that a physical association exists between them. By combining co-recruitment analysis of the time-resolved and quantitative CBL and CBLB signalosomes with biochemical analysis, we demonstrated that CD5, a transmembrane protein expressed at the surface of T cells, constitutes a scaffold that has a central role in the CBL- and CBLB-mediated ubiquitylation events that follow TCR engagement.

## Results

### Gene-targeted mice suitable for proteomics analysis of CBL and CBLB and determination of their relative cellular abundance

To identify by AP-MS the proteins interacting with CBL and CBLB prior to or after TCR-mediated activation of primary mouse T cells,

we generated two lines of gene-targeted mice expressing One-STrEP-tag (OST) (Junttila *et al*, 2005) at the carboxyl-terminus of endogenous CBL and CBLB proteins (Fig EV1A and B). Analysis of mice homozygous for the *Cbl*$^{OST}$ allele (also known as *B6-Cbl*$^{tm1Mal}$; called "CBL$^{OST}$" mice here) and for the *Cblb*$^{OST}$ allele (also known as *B6-Cblb*$^{tm1Ciphe}$; called "CBLB$^{OST}$" mice here) showed that their T cells developed properly and that they contained normal numbers of CD4$^+$ and CD8$^+$ T cells (Fig EV1C–F). CD4$^+$ T cells purified from wild-type, CBL$^{OST}$, and CBLB$^{OST}$ mice and stimulated with anti-CD3 antibodies in the presence or absence of anti-CD28 antibodies showed that expression of the CBL-OST and CBLB-OST proteins had no detrimental effect on their proliferation (Fig EV1G and H) or production of interleukin 2 (Fig EV1I and J). Immunoblot analysis showed that before and after activation with anti-CD3 and anti-CD4 antibodies, CBL-OST and CBLB-OST proteins can be purified from lysates of CBL$^{OST}$ and CBLB$^{OST}$ CD4$^+$ T cells, respectively, using Sepharose beads coupled to Strep-Tactin, a streptavidin derivative that binds with high affinity to the OST sequence (Fig 1A). As expected, no detectable material was recovered from wild-type CD4$^+$ T cells. Therefore, primary CD4$^+$ T cells from CBL$^{OST}$ and CBLB$^{OST}$ mice are normal and can be used in AP-MS analysis.

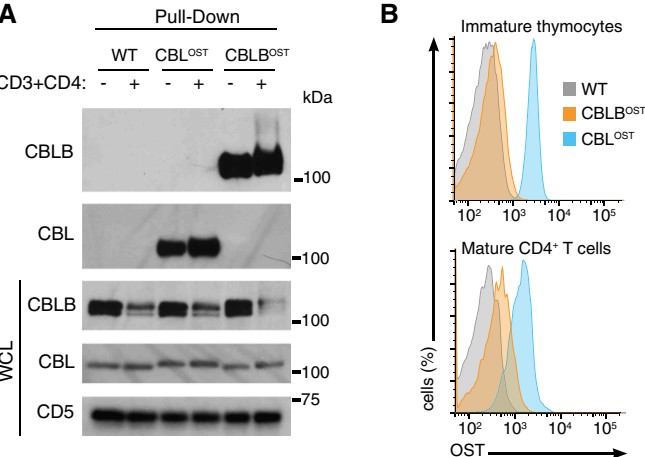

**Figure 1. Purification of CBL-OST and CBLB-OST proteins and quantification of their relative abundance.**

A   CD4$^+$ T cells from wild-type (WT), CBL$^{OST}$, and CBLB$^{OST}$ mice were left unstimulated (−) or stimulated for 2 min with anti-CD3 and anti-CD4 antibodies (+). Equal amounts of cell lysates were subjected to affinity purification on Strep-Tactin Sepharose beads, followed by elution of proteins with D-biotin. Eluted proteins were analyzed by immunoblot with antibodies specific for CBLB and CBL. Equal amounts of proteins from whole-cell lysates (WCLs) were also probed for CBLB and CBL. Note that in the case of CBLB, two bands are detected and likely corresponded to the presence of two isoforms differing at their N-terminus (http://www.uniprot.org/uniprot/Q3TTA7). Also shown are the loading control corresponding to the same immunoblot probed with anti-CD5, and molecular masses (kDa). Data are representative of at least three experiments.

B   CD4$^+$CD8$^+$ thymocytes and mature CD4$^+$ T cells isolated from mice with the specified genotypes were permeabilized and stained with saturating amounts of Strep-Tactin APC and analyzed by flow cytometry. Background staining was deduced using cells from WT mouse. Data are representative of two experiments.

The need to use distinct antibodies to detect CBL and CBLB precludes raising any conclusion on their relative abundance. Considering that CBL-OST and CBLB-OST proteins are expressed at levels comparable to that of their wild-type counterparts (Fig 1A), the shared OST can be used to measure their relative abundance using Strep-Tactin coupled to the allophycocyanin (APC) fluorescent probe. Intracytoplasmic staining of double-positive (CD4$^+$CD8$^+$) thymocytes—the major population of developing T cells found in the thymus—and of mature CD4$^+$ T cells with Strep-Tactin APC showed that CBL was less abundant in mature CD4$^+$ T cells than in double-positive thymocytes, whereas the opposite occurred for CBLB (Fig 1B). Consistent with a recent study of CD8$^+$ T cells (Hukelmann *et al*, 2016), we found that CBL was 3.8-fold more abundant than CBLB in mature CD4$^+$ T cells. Therefore, in contrast to previous assumptions (Naramura *et al*, 2002), CBL showed a greater cellular abundance than CBLB in both developing and mature T cells.

## Characterization of the CBL and CBLB signalosomes in primary CD4$^+$ T cells

To determine the composition of the CBL and CBLB signalosomes, we isolated CD4$^+$ T cells from CBL$^{OST}$ and CBLB$^{OST}$ mice, lysed them with non-ionic detergent before or at various times after activation with anti-CD3 and anti-CD4 antibodies, and then isolated protein complexes containing CBL-OST and CBLB-OST using Strep-Tactin. Three independent biological experiments each involving five different conditions corresponding to no stimulation and to four time points spanning 600 s after anti-CD3 and anti-CD4 stimulation were analyzed by AP-MS. The reproducibility of the AP-MS process was assessed for each condition of stimulation across technical and biological replicates (Fig EV2). To distinguish truly interacting proteins from non-specific contaminants, control AP-MS experiments were performed for each time point using wild-type CD4$^+$ T cells. Protein intensities were quantified using the MaxQuant software (Dataset EV1 and Data availability section) and normalized across the different conditions (see Materials and Methods). To determine whether a given detected protein was specifically associated with the CBL-OST (or CBLB-OST) bait over the course of an experiment, we compared for each time point the distribution of log-normalized intensities obtained for CBL$^{OST}$ (or CBLB$^{OST}$) and for wild-type CD4$^+$ T cells. Each comparison yielded a value denoted as $r(t)$ and specifying the enrichment observed in CBL$^{OST}$ (or CBLB$^{OST}$) CD4$^+$ T cells as compared to wild-type CD4$^+$ T cells and a corresponding *P*-value based on a one-way ANOVA test (denoted as $P(t)$; Fig 2A). At a given time $t$, proteins were selected as specific interacting partners of the considered OST bait when both the *P*-value $P(t)$ was below a set threshold and the corresponding enrichment $r(t)$ was greater than twofold (see Materials and Methods). To further avoid adventitious inclusion of non-specific partners, we repeated this selection process 200 times by resampling the experimental intensities independently from the original distribution using a bootstrap algorithm. Only proteins that were selected as specific partners in at least 90% of such tests were kept (Fig 2B), resulting in 98 and 43 unique proteins specifically interacting with CBL and CBLB, respectively, in at least one condition of stimulation (Fig 2C and D). The specificity of our approach can be illustrated by the CBL-OST bait, the normalized intensity of which was at least

two orders of magnitude higher in CBL$^{OST}$ CD4$^+$ T cells than in wild-type CD4$^+$ T cells (Fig 2A and B).

Whereas a few interacting proteins (preys) such as UBASH3A (also known as STS-2) were found constitutively associated with the CBL-OST and CBLB-OST bait, most preys showed a transient pattern of interaction (Fig 2E). In the case of the CBL signalosome, the number of interactors varied non-monotonously as a function of time, a peak of 63 interactors being detected 30 s after TCR stimulation followed by a second peak of smaller complexity 10 min after TCR stimulation. In contrast, the CBLB signalosome reached its greatest complexity 2–5 min after TCR stimulation (Fig 2E). Therefore, the kinetics of assembly with and disassembly of the preys from the CBL and CBLB baits after TCR stimulation are not superimposable. 33 and 14% of the proteins identified as CBL and CBLB interactors, respectively, have been already reported in the BioGRID database (Fig 2F). Such overlap corresponded to interacting proteins that in our study ranked among those with the highest maximum mean enrichment (Fig 2C and D). Therefore, our approach provided a more comprehensive view of the CBL and CBLB signalosomes in that it encompasses a majority of interacting proteins not found the BioGRID database.

## Proteins interacting with both CBL and CBLB

As anticipated from the structural similarities existing between CBL and CBLB (Lee & Tsygankov, 2013), 15 proteins were capable of binding to both CBL and CBLB (Fig 3). UBASH3A and UBASH3B (also known as STS-1) belonged to these shared interactors and use their SH3 domain to bind to the proline-rich region of CBL and CBLB (Feshchenko *et al*, 2004). UBASH3A and UBASH3B comprise a histidine phosphatase domain capable of dephosphorylating phosphotyrosine-containing substrates. UBASH3B has, however, a stronger tyrosine phosphatase activity than UBASH3A and by dephosphorylating ZAP-70 acts as a negative regulator of TCR signaling (San Luis *et al*, 2011; Luis & Carpino, 2014; Yang *et al*, 2015). CSK and CD5, two negative regulators of TCR signaling, also interacted with both CBL and CBLB. The PTK CSK stabilizes SRC-family PTKs in their inactive configuration, and CD5 is a transmembrane receptor that attenuates signals arising from TCR engagement (Tarakhovsky *et al*, 1995; Azzam *et al*, 2001). Adaptor proteins belonging to the GRB2 (GRB2, GRAP, and GRAP2) and CRK families (CRK and CRKL) were also present in both signalosomes. CRK family members are thought to enhance the binding of CBL to PI3K (Gelkop *et al*, 2001), whereas members of the GRB2 family connect CBL and CBLB to the TCR signal transduction network (Buday *et al*, 1996). Upon CBL- and CBLB-mediated ubiquitylation, elements of this signaling network are then recognized by components of the endocytic machinery, and two of them, ITSN2 and EPS15L1 (McGavin *et al*, 2001; Haglund & Dikic, 2012), stood among the interacting partners common to CBL and CBLB. Although no E2 ubiquitin-conjugating enzyme was recovered in the CBL and CBLB signalosomes, ubiquitin constituted one of the species common to both signalosomes. Ubiquitin was 4.7 times more enriched in the CBLB signalosome ($<r>_{max}$ = 16.9) than in that of CBL ($<r>_{max}$ = 3.57). Congruent with the role played by ubiquitin in signal transduction, endosome sorting, and protein degradation, when the CBL and CBLB signalosomes were subjected to GO term enrichment analysis (Bindea *et al*, 2009), a significant

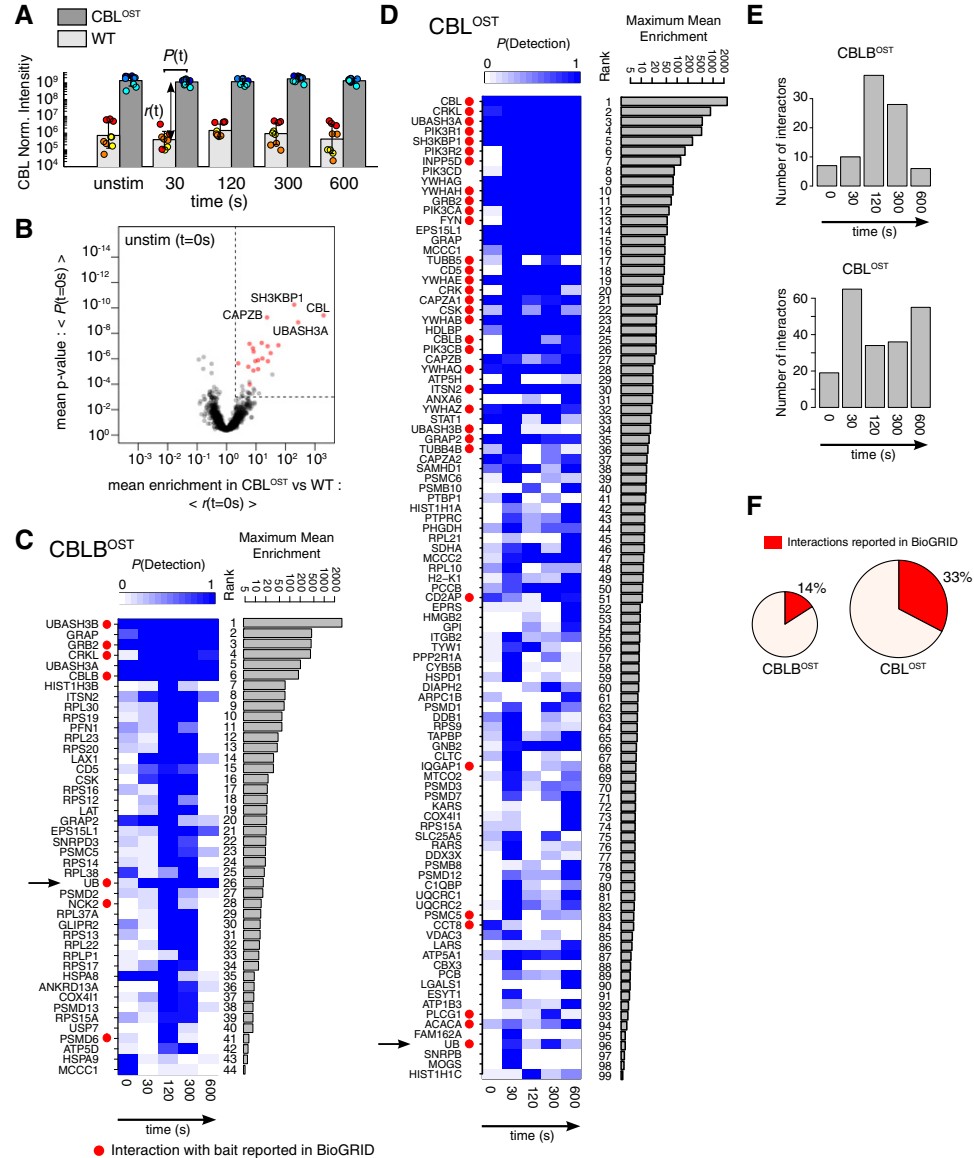

**Figure 2. Detection of specific interacting partners of CBL and CBLB in peripheral CD4$^+$ T cells prior to and following TCR stimulation.**

CD4$^+$ T cells from wild-type (WT), CBL$^{OST}$, and CBLB$^{OST}$ mice were left unstimulated (unstim) or stimulated for 30, 120, 300, or 600 s with anti-CD3 and anti-CD4 antibodies. Equal amounts of cell lysates were subjected to affinity purification using Strep-Tactin Sepharose beads and then to MS analysis as described in Materials and Methods.

A   Histogram comparing the normalized CBL protein intensities resulting from AP-MS analysis of CD4$^+$ T cells from WT and CBL$^{OST}$ mice prior to and after the specified stimulation times. For each time point, the distributions of the normalized intensities of the nine data points (three biological replicates × three technical replicates) corresponding to CD4$^+$ T cells from wild-type (WT) and CBL$^{OST}$ mice were compared using the adjusted $P$-value from a one-way ANOVA test (denoted as $P(t)$) and the value $r(t)$ corresponding to the enrichment observed in the CBL$^{OST}$ samples as compared to WT samples. Different colors represent distinct biological replicates.

B   Proteins are classified as CBL interactors according to their position in a volcano plot in which the mean $P$-value $<P(t)>$ is plotted against the corresponding mean enrichment $<r(t)>$ for CBL$^{OST}$ versus WT samples. The time point shown corresponds to the unstimulated condition ($t = 0$ s). Mean was calculated using bootstrap resampling (see Results and Materials and Methods). Proteins that displayed an enrichment $r(t)$ greater than twofold and a corresponding $P$-value $P(t)$ lower than a set threshold (see Materials and Methods) in more than 90% of the bootstrap iterations were selected as specific partners (indicated in red). Names of four of the most significantly enriched proteins are indicated. Dashed lines represent thresholds on the $P$-value $P* = 10^{-3}$ and the enrichment $r* = 2$ used to identify specific interactors of CBL.

C, D List of specific interacting partners of CBLB (C) and CBL (D) ranked according to their maximum mean enrichment across all time points. Heat maps show, for each time point, the fraction of bootstrap iterations for which the corresponding proteins were detected as specific partners (denoted as "$P$(Detection)"). Red dots indicate the proteins that have been previously reported as CBLB (C) or CBL (D) interactors in the BioGRID database. Ubiquitin (UB) is highlighted with a black arrow.

E   Histograms showing the numbers of specific interactors binding to the CBL-OST or CBLB-OST proteins prior to and after TCR stimulation for different times and for which $P$(Detection) > 0.9.

F   Pie charts showing the percentages of CBL and CBLB interactors detected in both the present study and the BioGRID public database. The size of the pie charts is proportional to the numbers of interactors per bait.

overrepresentation of annotations related to "signaling adaptor activity" and "role in protein degradation" was observed (Fig EV3). In line with studies performed in adipocytes and fibroblasts and suggesting that CBL and CBLB are capable of heterodimerizing (Liu *et al*, 2003; Rorsman *et al*, 2016), CBLB was found in the CBL signalosome (Fig 2D). The lower cellular abundance of CBLB as compared to CBL (Fig 1B) and the detection limit achieved in our AP-MS experiments might explain why no CBL-specific peptide was recovered from the CBLB signalosome (Fig 2C).

### Proteins specifically interacting with CBL or CBLB

Some GO terms ("phosphatidylinositol complex", "F-actin capping protein complex") were enriched only in the CBL signalosome, whereas others ("cytosolic ribosome") were more enriched in the CBLB signalosome, highlighting that CBL and CBLB each control the formation of a unique signalosome. In support of that view, CBLB but not CBL associated with the transmembrane adaptors LAT and LAX1. LAT is essential for TCR-mediated T-cell activation, whereas LAX1 is thought to negatively regulate T-cell activation (Horejsi *et al*, 2004). Out of the six baits (ZAP-70, LAT, SLP-76, PAG-CBP, CBL, CBLB) analyzed to date using TCR-activated primary CD4[+] T cells and a OST-based approach (Roncagalli *et al*, 2014; Reginald *et al*, 2015), CBLB was the only bait capable of precipitating a large number of components of the small and large ribosomal subunits following TCR activation (Figs 3 and EV3). Ubiquitin is encoded by four different genes that include the *Uba52* and *Rps27a* genes that code for a single copy of ubiquitin fused to the ribosomal proteins RPL40 and RPS27A, respectively. Interestingly, in some species,

ubiquitin remains fused to RPS27A once it is incorporated into the mature ribosome (Catic *et al*, 2007). The identification of one peptide specific for RPS27A in the CBLB signalosome (indexed under ubiquitin) suggests that an interaction between the UBA of CBLB and RPS27A may account for the pull-down of the 14 ribosomal components that are uniquely recruited by CBLB following T-cell activation. Alternatively, the recovery of such numerous ribosomal components in the CBLB signalosome might be due to ubiquitylation of ribosomal proteins or of ribosome-associated chaperones by ubiquitin-protein ligases of the RING family (Panasenko, 2014). Whether CBLB plays a role in ribosomal biology, however, remains to be determined.

Among the large number of interactors specific to the CBL signalosome (Fig 3) were several proteins involved in phospho-inoside-based signaling (such as PLCG1, INPP5D, and several regulatory (R1 and R2) and catalytic (CA, CB, and CD) subunits of PI3K), six members of the 14-3-3 protein family, and proteins involved in clathrin-mediated endocytosis (CLTC, CD2AP, or SH3KBP1) and in cytoskeletal rearrangement (such as the F-actin capping proteins CAPZA1 and CAPZA2, the tubulins TUBB4B and TUBB5, the Diaphanous-related formin DIAPH2, and the chaperone CCT8). FYN, a member of the SRC-family PTKs, interacted specifically with CBL but not CBLB, echoing its role in the recruitment of PI3K to CBL upon phosphorylation of the tyrosine residue found at position 731 (Hunter *et al*, 1999). Therefore, our analysis provided a comprehensive view of the CBL and CBLB signalosomes after TCR stimulation of primary CD4[+] T cells, highlighting both the redundant and unique functional features of these two E3 ubiquitin–protein ligases and identifying 66 and 37 proteins not

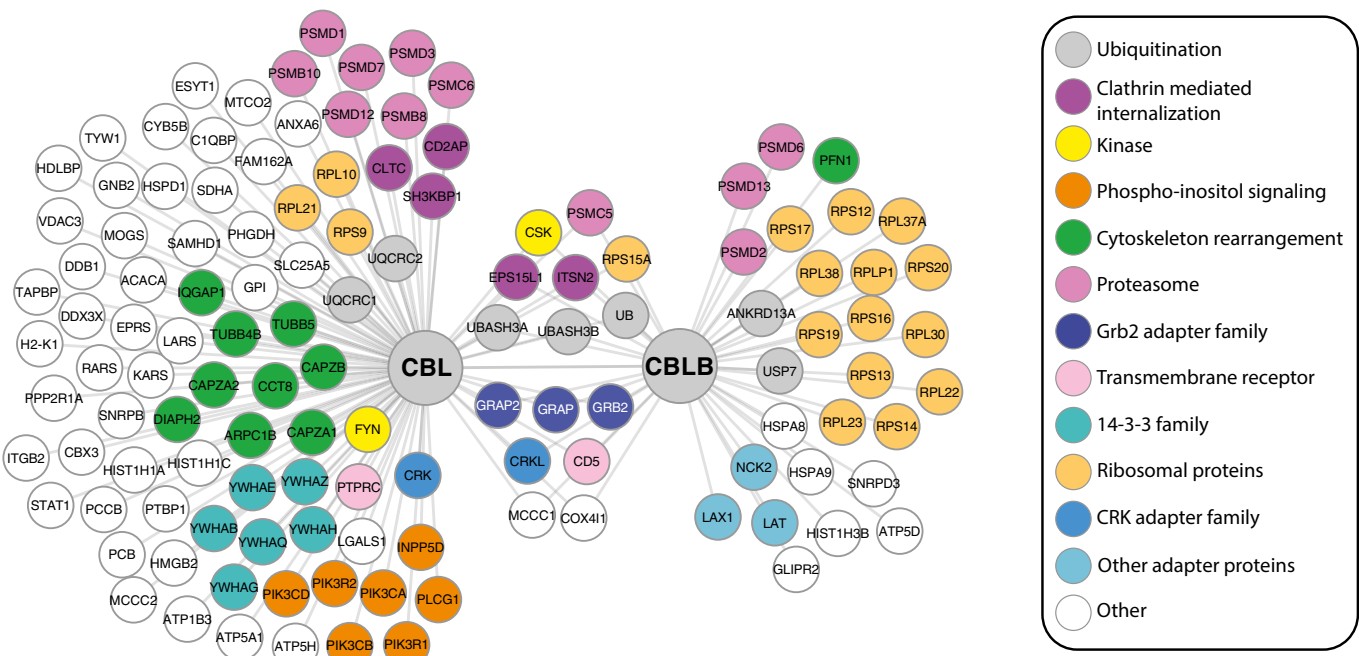

**Figure 3.   Comparison of the CBL and CBLB signalosomes of peripheral CD4[+] T cells.**
Each protein interacting with either or both the CBL-OST and CBLB-OST baits is represented as a node and is linked by an edge to the corresponding bait. Proteins are identified by their BioGRID designations and color-coded according to their function or protein family; see key (right). As specified, a set of proteins interacts with both CBL and CBLB, and CBLB is found in the CBL signalosome.

found associated with CBL and CBLB, respectively, in the BioGRID database.

## Co-recruitment analysis of time-resolved CBL and CBLB signalosomes

To attempt to decipher the underlying PPIs responsible for the assembly of the CBL and CBLB signalosomes, we further analyzed the kinetics of formation of such signalosomes. We reasoned that if two recruited proteins are interacting with each other or are part of a macromolecular complex, their kinetics of recruitment over the first 600 s of TCR signaling should be similar. To quantify the similarity in the recruitment kinetics of a given pair of proteins, we used the Pearson correlation coefficient (see Materials and Methods and Fig 4A). We then carried out this computation for each pair of recruited proteins, and we partitioned the resulting correlation matrix into distinct clusters using a standard K-means algorithm (Fig EV4). In the case of the CBL signalosome, this partitioning revealed that proteins that are structurally related (for instance, 14-3-3 family members or proteins containing a "capping protein interaction" motif (CAPZA1, CAPZA2, CD2AP, and SH3KBP1)) or functionally connected (for instance, those involved in phosphoinositide-based signaling) were grouped within the same cluster (Fig EV4).

To further refine the analysis of correlations in protein recruitment within the CBL and CBLB signalosomes and explore the finer structure of the correlation matrix, we constructed the corresponding co-recruitment network (CN) (Fig 4B and C). In CN, nodes represent specific interacting partners and edges connect pairs of nodes that were co-recruited and for which the *P*-value associated with the Pearson correlation coefficient exceeded a set threshold (see Materials and Methods). As expected from the correlation matrix, the CNs of the CBL and CBLB signalosomes were organized into highly connected subnetworks corresponding to proteins belonging to the same structural family (for instance, the six identified members of the 14-3-3 family proteins or the GRB2 and GRAP adaptors), or involved in a given intracellular process (such as phosphoinositide signaling, clathrin-mediated endocytosis, or cytoskeletal rearrangement) or a macromolecular complex (the proteasome) (Fig 4B and C).

## Predicting PPIs using co-recruitment analysis

To test whether the correlations observed in the timing of co-recruitment could be predictive of actual physical interactions between co-recruited preys, we compared the interactions deduced from our CN analysis to the whole set of CBL- and CBLB-interacting partners reported in the BioGRID database. Among the 328 interactions predicted to exist on the basis of the CBL CN, 62 were already reported as physical interactions within the *Homo sapiens* and *Mus musculus* BioGRID databases (Fig 4D). The probability of accurately predicting an existing interaction was thus 0.19 ($P = 62/328$), a value largely exceeding the probability of 0.06 ($P = 302 \times 2/(98 \times 97)$) of finding one of the 302 reported interaction among $N \times (N-1)/2$ possible interactions in a random network comprising $N = 98$ nodes. Moreover, the probability to properly predict an existing interaction reached 0.31 ($P = 37/120$) when only edges with a Pearson correlation coefficient greater than 0.8 were considered (Fig 4D). Likewise, in the case of the CBLB CN, 7 out of 34

($P = 0.21$) of the predicted interactions were reported in BioGRID and 2 out of 8 ($P = 0.25$) remained when the network was restricted to edges with $R > 0.8$ (Fig 4E), thereby exceeding the odds to accurately predict a reported interaction in a random network with the same number of nodes ($P = 0.15$). Therefore, the correlations that exist among some preys in their kinetics of assembly with and disassembly from the CBL and CBLB baits over 600 s of TCR stimulation can be used as a decision support tool to infer candidate PPIs.

## Validation of novel PPIs predicted on the basis of the CBL and CBLB correlation networks

To further test the predictive power of our CN analysis, we took advantage of the fact that CBLB was present in the CBL signalosome (Fig 4B). We focused on the CBLB first neighbors found in the CBL CN and posited that they should be detected as interacting partners of CBLB in the experimentally determined CBLB signalosome (Fig 3). Indeed, 5 out of the 13 (38.5%) CBLB first neighbors found in the CBL CN constituted interacting partners of CBLB (Fig 5A). This compared favorably with the chance (15.3%) to detect one of the 15 CBLB-specific interactors found within the CBL signalosome (Fig 3) if edges were drawn at random. In addition, the percentage of identified CBLB interactors increased to 66.7% when focusing on the sole edges with a Pearson correlation coefficient $R > 0.8$. We further observed that the percentage of CBL interactors detected as specific partners of CBLB rapidly decreased as a function of their distance to CBLB within the CBL CN (Fig 5B). This holds true also in the high-confidence CBL CN in which only edges with $R > 0.8$ were considered. Therefore, the presence of an edge in the CBL CN was predictive of the probability that a direct physical association exists between the linked nodes.

Based on the above observation, we used the CBL CN to construct the high-confidence first-neighbors subnetwork corresponding to CRKL and CSK and found that only 4 out of the 19 predicted interactions were previously reported in BioGRID (Fig 5C). Within this subnetwork, the interaction between CSK and CD5 was both novel and exhibited the highest correlation ($R = 0.96$). To confirm that this interaction occurred in stimulated T cells, coimmunoprecipitation of proteins from cell lysates of primary CD4$^+$ T cells was performed before and after TCR stimulation. It showed that CSK was capable of interacting with CD5 upon TCR stimulation (Fig 5D). Additional coimmunoprecipitations confirmed predicted PPIs involving CSK-CBLB, CRKL-PI3Kp85β, CSK-PI3Kp85β, PI3Kp85β-PI3Kp110α, CSK-PI3Kp110α, and CRKL-PI3Kp110α (Fig 5D–G), whereas the CBLB-CD5 PPI was identified in the CBLB signalosome (Fig 2C). Conversely, we failed validating predicted PPIs involving CRKL-CSK and PI3Kp85β-CD5. Therefore, among the 11 PPIs interactions that can be tracked by coimmunoprecipitation analysis due to the existence of specific antibodies, 9 were validated by coimmunoprecipitation (Fig 5H). False positives are expected in CN derived from such dense correlation matrix and alternative methods such as Gaussian Graphical Models or Bayesian Network can likely be used to discard potential non-direct effects (Pe'er, 2005; Krumsiek *et al*, 2011). Our results indicated that increasing the threshold on the Pearson $R$ coefficient reduced the fraction of such false positives (Fig 5A and B). Therefore, our results demonstrate that CN can predict novel physical associations between the preys recruited by a given bait.

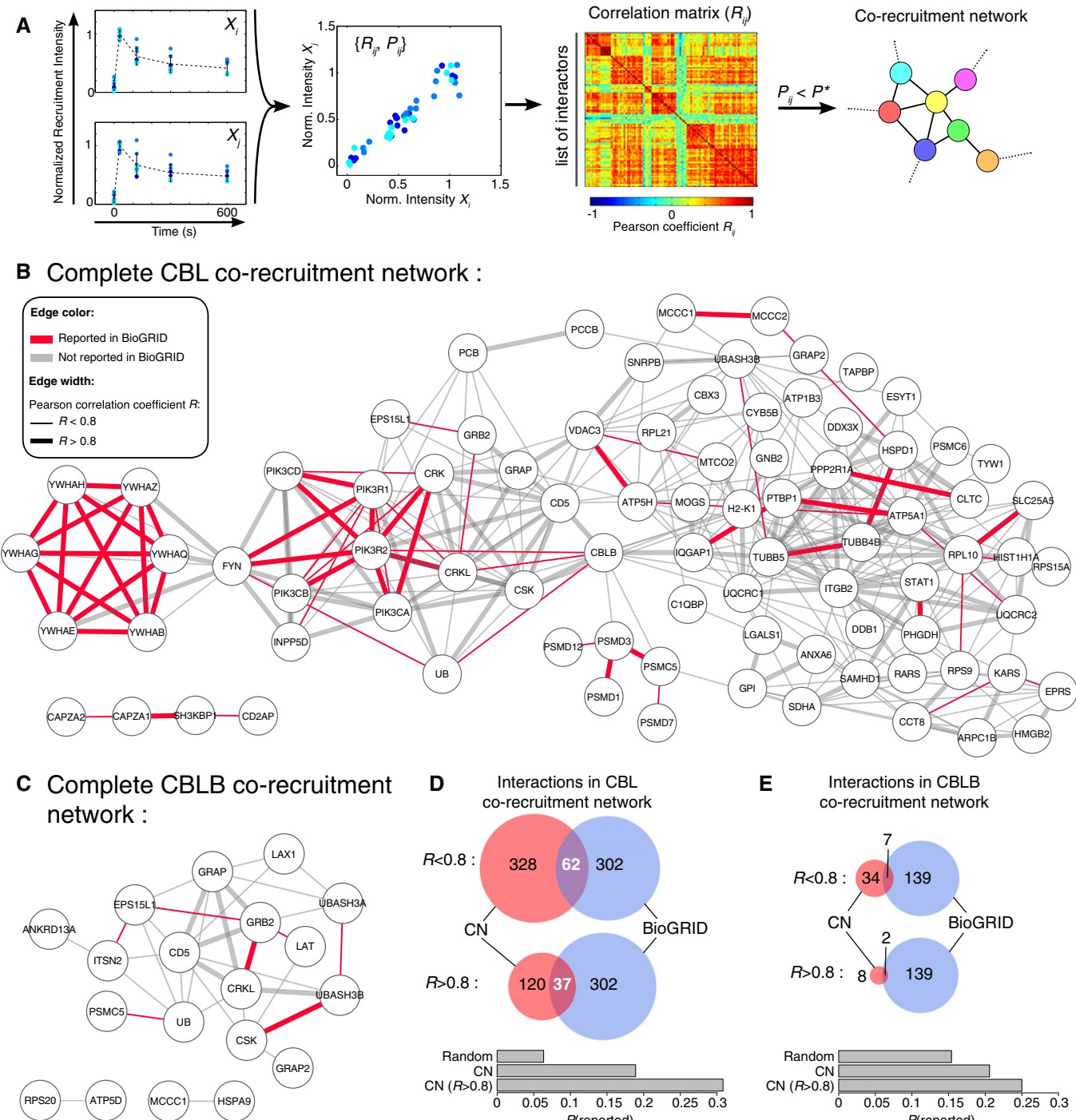

**Figure 4. Analysis of the correlations existing in the assembly with and disassembly from the CBL-OST or CBLB-OST baits for all pairs of identified preys.**

A    Workflow used for the generation of the co-recruitment correlation network (CN). Left: Normalized intensities of recruitment of two identified partners (denoted $X_i$ and $X_j$) to the considered bait as a function of time. Center left: Normalized intensities of recruitment for preys $X_i$ and $X_j$ are plotted against each other ($R_{ij}$: Pearson correlation value; $P_{ij}$ corresponding $P$-value). Center right: Correlation matrix ($R_{ij}$) between all pairs of identified interactors partitioned into different clusters. Right: Portion of the obtained correlation network after filtering according to $P_{ij}$ values.

B, C    Representation of the CBL (B) and CBLB (C) CNs. Nodes represent specific interacting partners and edges connect pairs of nodes for which the $P$-value associated with the Pearson correlation coefficient exceeded a set threshold. Edge width is coded according to Pearson correlation coefficient ($R < 0.8$ or $R > 0.8$). The identified edges are depicted in red if they have been already reported as direct interaction partners in the BioGRID database.

D, E    Venn diagrams showing overlaps between protein–protein interactions deduced from the CBL (D) or CBLB (E) CNs and those reported in the BioGRID database for *Homo sapiens* and *Mus musculus*. Two sets of analysis are shown according to Pearson correlation coefficient values ($R < 0.8$ or $R > 0.8$). Histograms below the Venn diagrams compare the probabilities to predict existing interactions with a random network (Random), the complete CN, or the high-confidence CN ($R > 0.8$).

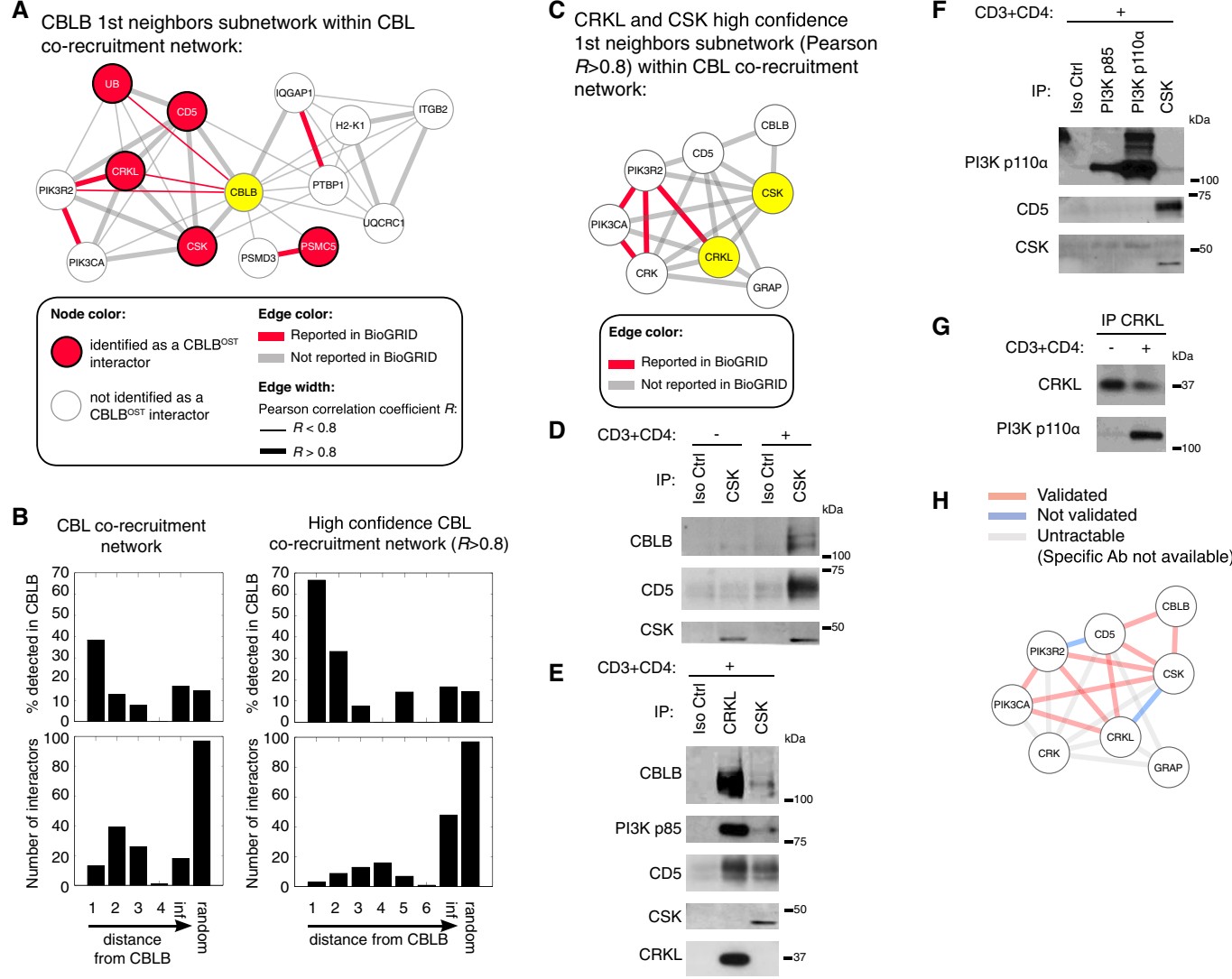

**Figure 5.  Co-recruitment network predicts the occurrence of physical association between pairs of recruited proteins.**

A    CBLB first-neighbors network within the CBL CN. Edge width is coded according to Pearson correlation coefficient ($R < 0.8$ or $R > 0.8$). The identified edges are depicted in red if they have been already reported as direct interaction partners in the BioGRID database. Proteins detected in the CBLB signalosome shown in Fig 3 are colored in red.

B    Histograms showing the number and corresponding percentage of CBL interactors also present in the CBLB signalosome according to their distance from CBLB in the CBL CN. The two depicted analyses correspond to the complete and to the high-confidence ($R > 0.8$) CNs.

C    Zoom on the high-confidence ($R > 0.8$) CRKL and CSK first-neighbors subnetwork found within the CBL CN. The identified edges are depicted in red if they have been already reported as direct interaction in the BioGRID database. CSK and CRKL are highlighted in yellow.

D–G   Experimental validation of the PPIs predicted on the basis of the high-confidence CN shown in (C). CD4[+] T cells were left unstimulated (−) or stimulated (+) with anti-CD3 and anti-CD4 antibodies and subsequently lysed. Equal amounts of cell lysates were incubated with isotype control (Iso Ctrl) or the specified antibodies and the resulting immunoprecipitates (IP) analyzed by immunoblot with the antibodies specified in the left margin. The validated PPIs predicted in (C) correspond to CSK-CBLB and CSK-CD5 (D), CRKL-PI3Kp85β and CSK-PI3Kp85β (E), PI3Kp85β-PI3Kp110α and CSK-PI3Kp110α (F), and CRKL-PI3Kp110α (G). Molecular masses are shown (kDa). Data are representative of at least two experiments.

H    Summary of the PPIs found in the high-confidence CSK and CRKL first-neighbors subnetwork (see C) that were validated by coimmunoprecipitation (see D–G). Interactions validated and not validated by coimmunoprecipitation are shown using red and blue edges, respectively. Interactions untractable by coimmunoprecipitation analysis due to the lack of specific antibodies are shown as gray edges.

Data information: In (C) and (H), PI3Kp85β and PI3Kp110α are denoted as PIK3R2 and PIK3CA, respectively.

## Functional interdependence between CBL and CBLB

CBL and CBLB are more than just E3 ubiquitin–protein ligases and constitute scaffolding proteins. As a result, immunoblot analysis of

lysates of CD4[+] T cells showed that upon TCR stimulation, several ubiquitylated proteins were associated to the CBLB-OST and CBL-OST baits (Fig 6A). Consistent with the higher enrichment of ubiquitin observed in the CBLB signalosome as compared to the CBL

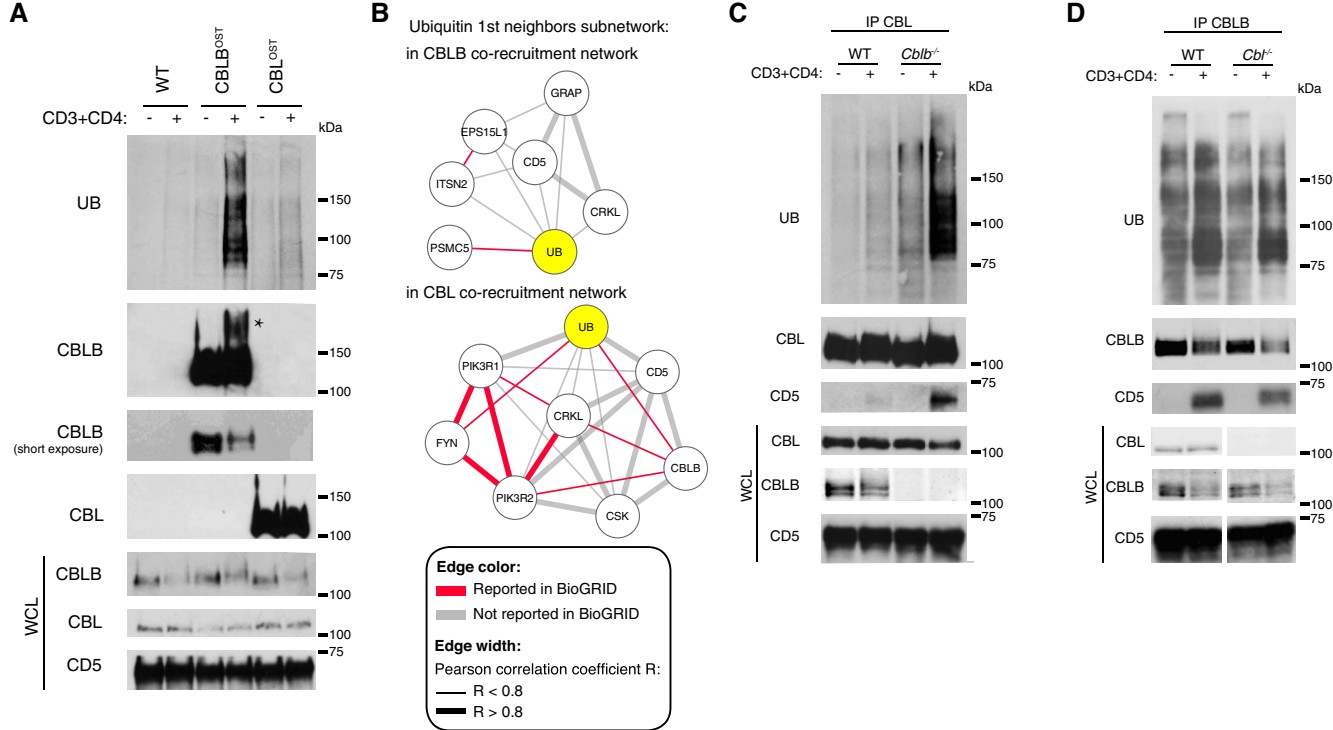

**Figure 6. Asymmetric functional relationships between CBL and CBLB.**

A CD4[+] T cells from wild-type (WT), CBL[OST], and CBLB[OST] mice were left unstimulated (−) or stimulated for 1 min with anti-CD3 and anti-CD4 antibodies (+). Equal amounts of cell lysates were subjected to affinity purification on Strep-Tactin Sepharose beads, and the resulting protein eluates were analyzed by immunoblotting with anti-ubiquitin (UB), anti-CBLB, or anti-CBL antibodies. Also shown are a loading control corresponding to equal amounts of proteins from whole-cell lysates (WCLs) probed with anti-CD5, and molecular masses (kDa). TCR stimulation resulted in some CBLB degradation and in the detection of a higher molecular weight CBLB form, likely due to its ubiquitylation (marked with an asterisk). In contrast, no degradation of CBL was detectable upon TCR stimulation and its molecular weight remained unchanged. Data are representative of at least two experiments.

B Representation of the ubiquitin (UB) first-neighbors subnetwork within the CBL and CBLB CN. Edge width is coded according to Pearson correlation coefficient ($R < 0.8$ or $R > 0.8$). Red edges depict direct interactions reported in the BioGRID database. Ubiquitin (UB) is highlighted in yellow.

C Immunoprecipitation of CBL from cell lysates of CD4[+] T cells isolated from WT or CBLB-deficient mouse ($Cblb^{-/-}$) and kept unstimulated (−) or stimulated as in (A). Immunoprecipitates and whole-cell lysates (WCLs) were analyzed by immunoblot with antibodies specific for the proteins specified in the left margin. Molecular masses are shown (kDa). Data are representative of at least two experiments.

D Immunoprecipitation of CBLB from cell lysates of CD4[+] T cells isolated from WT or CBL-deficient mouse ($Cbl^{-/-}$) and kept unstimulated (−) or stimulated as in (A). Immunoprecipitates and whole-cell lysates (WCLs) were analyzed by immunoblot with antibodies specific for the proteins specified in the left margin. Molecular masses are shown (kDa). Data are representative of at least three experiments.

signalosome (see above), a stronger association was detected between ubiquitylated proteins and the CBLB-OST bait relative to the CBL-OST bait (Fig 6A). To gain further insights on the regulation of CBL- and CBLB-mediated ubiquitylation following TCR engagement, we constructed the first-neighbors subnetwork of ubiquitin in the CBL and CBLB CNs (Fig 6B). CBLB is part of the CBL signalosome (Fig 3) and constituted a first neighbor of ubiquitin in the CBL CN. CBLB and ubiquitin showed a Pearson correlation coefficient $R = 0.74$, suggesting that CBLB recruitment to CBL correlated with the onset of ubiquitylation in the CBL signalosome. To analyze whether the presence of CBLB within the CBL signalosome contributed to the presence of ubiquitin species, we performed immunoprecipitation of CBL from lysates of TCR-stimulated CD4[+] T cells isolated from CBLB-deficient mice. Unexpectedly, the lack of CBLB dramatically enhanced the association between CBL and ubiquitylated species (Fig 6C). In contrast, the lack of CBL did not change the level of ubiquitylated species found within the CBLB complexes

(Fig 6D). Therefore, although asymmetric, a functional interdependence exists between CBL and CBLB in CD4[+] T cells and controls the capacity of CBL to recruit ubiquitylated proteins following TCR stimulation. Plausible mechanisms for such an unexpected finding are discussed below.

## CD5 regulates CBL- and CBLB-mediated ubiquitylation following TCR activation

Our AP-MS approach revealed that CD5 is the only transmembrane receptor expressed by CD4[+] T cells capable of associating with both CBL and CBLB after TCR stimulation (Fig 3), a finding confirmed using coimmunoprecipitation (Figs 6C and D, and 7A). Interestingly, the presence of CD5 was highly correlated with that of ubiquitin in both the CBL and CBLB CNs (Pearson correlation coefficient of 0.83 and 0.77, respectively; Fig 6B). Accordingly, we anticipated that CD5 could modulate the recruitment of ubiquitylated proteins

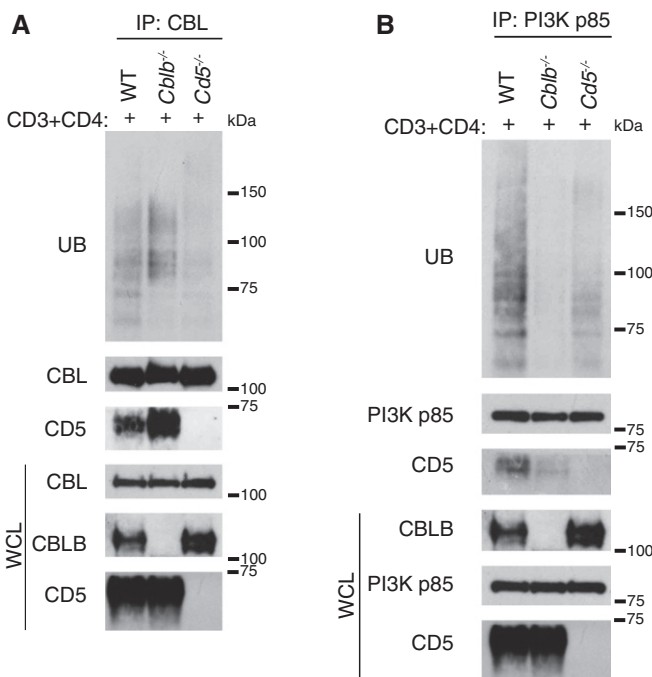

**Figure 7.  CD5 regulates CBL-mediated ubiquitylation following TCR stimulation in mature CD4⁺ T cells.**

A, B  CD4⁺ T cells from wild-type, *Cblb*−/−, and *Cd5*−/− mice were stimulated for 1 min with anti-CD3 and anti-CD4 antibodies (+). CBL (A) and PI3KR1 (PI3Kp85α) (B) were immunoprecipitated from equal amounts of protein lysates. Level of ubiquitylation and amount of CD5 associated proteins were evaluated by immunoblot with antibodies specific for the proteins specified in the left margin. Whole-cell lysates (WCLs) were also analyzed with the specified antibodies. Molecular masses are shown (kDa). Data are representative of at least two experiments.

to CBL or CBLB. Consistent with that view, immunoprecipitation of CBL from lysates of TCR-stimulated CD4⁺ T cells isolated from CD5-deficient mice showed that the absence of CD5 resulted in a drastic reduction in the extent of ubiquitylated proteins that coimmunoprecipitated with CBL as compared to TCR-stimulated wild-type CD4⁺ T cells (Fig 7A). PIK3R1 (also known as PI3K p85α) is found in the sole CBL signalosome and akin to CD5 constituted a first neighbor of ubiquitin within the CBL CN (Pearson correlation coefficient of 0.82; Fig 6B). Ubiquitylated proteins coimmunoprecipitated with PI3KR1 in TCR-stimulated wild-type CD4⁺ T cells, and their presence was reduced in the absence of CBLB, in line with a former study (Fang & Liu, 2001), and of CD5 (Fig 7B). Altogether, these results demonstrate that CD5 plays an important role in CBL- and CBLB-mediated ubiquitylation in CD4⁺ T cells following TCR engagement.

## Discussion

In this study, we combined mouse genetics, time-resolved AP-MS and computational approaches to investigate the composition, dynamics, and internal organization of the CBL and CBLB signalosomes after TCR stimulation of primary CD4⁺ T cells. As expected

from the structural similarity existing between CBL and CBLB, both signalosomes shared a number of interacting partners. However, these shared partners represent only 15 out of the 117 unique proteins that are enumerated when the CBL and CBLB signalosomes are pooled together. In addition, the two signalosomes show distinct kinetics of assembly. As discussed below, this may account for the fact that CBL and CBLB exert both redundant and unique functions in mature CD4⁺ T cells. Most of the interactors identified in this study have not been previously implicated in the CBL and CBLB signalosomes, and they constitute a valuable resource to further our understanding of the role of CBL and CBLB in the TCR signal-transduction network.

To go beyond the mere inventory of the CBL and CBLB signalosomes, we exploited correlations in protein association with the CBL and CBLB baits as a function of time of TCR stimulation. This permitted us to predict the occurrence of physical association between some of the recruited proteins and to gain insights on how CBL and CBLB regulate protein ubiquitylation following TCR stimulation. The negative role of CBL and CBLB in TCR signaling has been primarily assigned to their ubiquitin–protein ligase activity (Paolino *et al*, 2011) and, as expected, ubiquitin was present in both signalosomes. CBLB associated with more ubiquitylated species than CBL: this may reflect that the CBLB signalosome contains a larger spectrum of ubiquitylation substrates than the CBL signalosome or be due to the higher affinity for ubiquitin of the UBA of CBLB as compared to that of CBL (Davies *et al*, 2004; Zhou *et al*, 2008). Unexpectedly, the lack of CBLB increased the association between CBL and ubiquitylated species. When compared to CBL, CBLB has the unique ability to nucleate complexes around the LAT and LAX1 transmembrane adaptors. Therefore, in the absence of CBLB, LAT and LAX1 can plausibly recruit CBL, thereby constituting novel "entry points" allowing CBL to reach unconventional substrates. Regardless of the mechanisms at play, the impact of the CBLB deficiency on the recruitment of ubiquitylated proteins by CBL emphasizes the complex rewiring processes that can affect signaling networks in T cells deprived of a single gene product.

Our AP-MS analysis further revealed that CD5 is the only transmembrane receptor expressed by CD4⁺ T cells capable of associating with both CBL and CBLB following TCR stimulation. Based on our co-recruitment analysis, we predicted and validated experimentally that CD5 is an important organizer of CBL- and CBLB-mediated ubiquitylation following TCR stimulation. CD5 is a type I transmembrane glycoprotein composed of three extracellular scavenger receptor cysteine-rich domains and a highly conserved cytoplasmic domain containing multiple serine, threonine, and tyrosine phosphorylation sites. Upon TCR stimulation, some of those tyrosine residues are phosphorylated with rapid kinetics, and in thymocytes and transformed T cells, they recruit several molecules including CBL (Dennehy *et al*, 1998; Lozano *et al*, 2000; Demydenko, 2010; Roa *et al*, 2013). The analysis of CD5-deficient mice established an inhibitory role for CD5 on TCR signaling (Tarakhovsky *et al*, 1995), and the negative regulation of TCR signaling by CD5 in T-cell hybridoma was shown to depend on the presence of the CD5 cytoplasmic domain (Pena-Rossi *et al*, 1999). Upon TCR engagement, CD5, CBL, and CBLB are rapidly recruited to the immunological synapse where they downregulate TCR signals (Brossard *et al*, 2003; Vardhana *et al*, 2010). Our study of primary T cells provides a rationale for such negative regulatory function in that it showed that upon

TCR-mediated phosphorylation, CD5 recruits CBL and CBLB at the inner face of the plasma membrane allowing ubiquitylation of several substrates (including themselves).

Elevated levels of TCR-CD3 complexes are expressed at the surface of CBL-deficient thymocytes and associated with enhanced tyrosine phosphorylation of LCK and ZAP-70, an observation accounted for by a lack of CBL-mediated ubiquitylation and subsequent degradation of the of TCR-CD3 complex (Wang *et al*, 2010). Moreover, CBLB deficiency reduces antigen-induced down-modulation of TCR expression at the surface of mature effector T cells (Shamim *et al*, 2007). However, neither TCR-CD3 complex subunits nor LCK and ZAP-70 were detectable in the CBL and CBLB signalosomes that assemble in mature T cells following TCR engagement. The only molecules involved in the regulation of proximal TCR signaling and present in both signalosomes were UBASH3B and CSK. UBASH3B reduces the level of TCR-induced ZAP-70 tyrosine phosphorylation (San Luis *et al*, 2011; Luis & Carpino, 2014; Yang *et al*, 2015), whereas CSK stabilizes LCK and FYN in their inactive configuration. It has been proposed that CSK negatively controls the catalytic activity of LCK via binding to the transmembrane adaptor PAG (also known as CBP) (Brdicka *et al*, 2000; Davidson *et al*, 2003). However, the lack of phenotype of PAG-deficient mouse and recent biochemical data suggest that additional proteins compensate for the loss of PAG and can recruit CSK at the inner face of the plasma membrane following TCR stimulation (Dobenecker *et al*, 2005; Xu *et al*, 2005; Reginald *et al*, 2015). Our AP-MS and co-recruitment analyses showed that CD5-CBL and CD5-CBLB complexes constitute one of these additional docking sites. Therefore, the negative role of the CD5-CBL and CD5-CBLB complexes in TCR signaling could be accounted for not only by their E3 protein ligase activity and capacity to recruit UBASH3B but also by their ability to recruit CSK as suggested by the influence of CD5 abundance on the phosphorylation of the regulatory tyrosine found at position 505 of LCK (Fig EV5). Analysis of Jurkat T cells showed that CD5 negatively regulates FYN (Bamberger *et al*, 2011), an observation likely due to the ability of CD5-CBL and CD5-CBLB complexes to bind to CSK as we have formally demonstrated here. Importantly, CBLB molecules unable to associate with E2 ubiquitin-conjugating enzymes fail to repress T-cell activation, suggesting that it is the absence of E3 ligase activity rather than the loss of other interactive domains within CBLB that causes hyperproliferation of CBLB-deficient mature T cells (Paolino *et al*, 2011). Therefore, in contrast to the non-redundant E3-ubiquitin ligase activity of CBLB, UBASH3B and CSK can plausibly use other scaffolding proteins to dampen TCR-proximal tyrosine kinases.

Mature T cells deprived of CBLB showed markedly increased responses to TCR stimulation as compared to wild-type T cells. Conversely, CBL-deficient mature T cells proliferate poorly in response to anti-CD3 stimulation. The interactors that are solely found in the CBLB signalosome may thus account for the negative regulatory role played by CBLB in mature T cells. Among them are the NCK2 adaptor, the small actin-binding protein PFN1 that regulates actin polymerization in response to extracellular signals, and ANKRD13A that binds to ubiquitylated species and regulates endocytosis of the epidermal growth factor receptor (Tanno *et al*, 2012) and of the B cell antigen receptor (Satpathy *et al*, 2015). The different "entry points" used by CBL and CBLB to dock at the plasma membrane and plug in the TCR signal transduction network might

also account for the contrasting regulatory functions exerted by CBL and CBLB in mature T cells. For instance, the LAT and LAX1 transmembrane adaptors are only found in the CBLB signalosome, and the binding of CBLB to LAT might uniquely attenuate TCR signaling via both LAT ubiquitylation (Balagopalan *et al*, 2011) and deactivation of TCR-proximal tyrosine kinases by UBASH3B and CSK.

The CBL signalosome differed from the CBLB signalosome by the presence of the receptor tyrosine phosphatase PTPRC and of the integrin β chain ITGB2. TCR stimulation also induced the association of CBL with six members of the 14-3-3 family proteins; 14-3-3 proteins are thought to modulate the interaction of CBL with signaling proteins (Liu *et al*, 1999). All the known catalytic subunits and two of the three regulatory subunits of class IA PI3K associated with CBL upon TCR stimulation. Binding of CBL to PI3K is mediated via the interaction of the SH2 domain of the p85 subunit of PI3K and the phosphorylated tyrosine found at position 731 of CBL, and it accounts for the effects exerted by CBL on cytoskeletal rearrangement (Lee & Tsygankov, 2013). This interaction localizes PI3K to the plasma membrane for the production of phosphatidylinositol (3,4,5)-trisphosphate (PIP3). After PI3K activation, increased PIP3 levels recruit PH domain-containing Rho-GEFs that activate downstream GTPases and in turn induce actin polymerization (Haglund & Dikic, 2012; Croise *et al*, 2014). Congruent with a role of a CBL-PI3K axis in the control of actin dynamics during endocytic processes, CBL associated with ARPC1B, a subunit of the Arp2/3 complex that is implicated in the control of actin polymerization, with IQGAP1, a scaffold protein involved in actin- and tubulin-based cytoskeletal reorganization that negatively regulates TCR stimulation (Gorman *et al*, 2012), with annexin A6 (ANXA6) that binds phospholipids in cellular membranes and contributes to regulation of endocytic transport, with the F-actin capping proteins CAPZA1 and CAPZA2, with the adaptor CD2AP and SH3KBP1, with the Diaphanous-related formin DIAPH2, with clathrin heavy chain (CLTC), and with tubulins TUBB4B and TUBB5. Association of the inositol polyphosphate-5-phosphatase INPP5D (also known as SHIP1) with CBL was also induced following T-cell activation, a paradoxical finding considering that INPP5D hydrolyzes PIP3 and thus opposes PI3K activity. The presence of PI3K and INNP5D in distinct CBL complexes might constitute a solution to this conundrum. Despite the fact that the signalosome of CBL is quantitatively richer than that of CBLB (our study), no detectable pathological condition resulted from the deletion of the *Cbl* gene (Naramura *et al*, 2002). The analysis of the *in vivo* consequences of constitutive gene inactivation has, however, clear limitations since mechanisms might be set in motion and capable of compensating the missing gene product. Conditional deletion of the *Cbl* gene in mature CD4[+] T cells will permit to obviate these limitations and to assess whether the unique features and richness of the CBL signalosome become functionally more blatant.

In conclusion, our study demonstrates the benefits of combining time-resolved AP-MS analysis with computational methods to exploit correlations in protein association with CBL and CBLB as a function of time of TCR stimulation for predicting the occurrence of physical association between them. By properly predicting novel PPIs among the CBL and CBLB interacting partners, we highlighted yet unappreciated mechanisms on how CBL and CBLB regulate

ubiquitylation following TCR stimulation. Finally, our work provides the basis for analyzing in a systematic and integrated manner the large numbers of interactomes that will be needed to make sense of the formidable complexity of the TCR signal-transduction network.

# Materials and Methods

### Construction of a OST-(Stop)$_2$-IRES2-mTFP1-loxP-frt-neo$^r$-frt cassette

A cassette containing a One-STrEP-tag (OST) sequence (Junttila *et al*, 2005), two STOP codons, an IRES2 sequence (derived from pIRES2-EGFP; Clontech), a Kozak sequence, and a sequence coding for mTFP1, a monomeric, bright, and photostable version of *Clavularia* cyan fluorescent protein (Ai *et al*, 2006), was abutted to the 5′ end of a frt-neo$^r$-frt cassette in which the neomycin–kanamycin resistance (neo$^r$) gene can be expressed under the control of a prokaryotic (gb2) or a eukaryotic (*Pgk1*) promoter.

### Cblb$^{OST}$ targeting vector

A 6.2-kb genomic fragment containing the 3′ end of the *Cblb* gene was isolated from a BAC clone (clone no. RP23-15M11; http://www.lifesciences.sourcebioscience.com) of C57BL/6J origin. Using recombination in *Escherichia coli*, a chloramphenicol-resistance gene bracketed at its 5′ end by a sequence that codes for a Gly-Ser-Gly spacer sequence comprising a *BspeE1* site and at its 3′ end by a *SalI* site was inserted in frame at the 3′ end of the *Cblb* coding sequence found in exon 19. Colonies containing a correctly inserted chloramphenicol resistance gene were selected, and the chloramphenicol resistance gene was excised using *BspeE1* and *SalI* digestion and replaced by a *XmaI-SalI* fragment corresponding to a OST-(Stop)$_2$-IRES2-mTFP1-loxP-frt-neo$^r$-frt cassette (see above). A loxP site was then introduced in the intron between exons 18 and 19. Finally, the targeting construct was abutted to a cassette coding for the diphtheria toxin fragment A (Soriano, 1997).

### Cbl$^{OST}$ targeting vector

A 7.5-kb genomic fragment containing exons 13–16 of the *Cbl* gene was isolated from a BAC clone (clone no. RP23-302P9; http://www.lifesciences.sourcebioscience) of C57BL/6J origin. A OST-(Stop)2loxP-tACE-CRE-PGK-gb2-neo$^r$-loxP cassette was introduced at the 3′ end of the *Cbl* coding sequence found in exon 16 as described (Roncagalli *et al*, 2014). Finally, the targeting construct was abutted to a cassette for the expression of thymidine kinase.

### Isolation of recombinant embryonic stem (ES) cell clones

Bruce 4 C57BL/6J ES cells (Kontgen *et al*, 1993) and JM8.F6 C57BL/6N ES cells (Pettitt *et al*, 2009) were electroporated with the *Cblb*$^{OST}$ and *Cbl*$^{OST}$ targeting vectors, respectively. After selection in G418 (*Cblb*$^{OST}$) or G418 and ganciclovir (*Cbl*$^{OST}$), ES cell clones were screened for proper homologous recombination by Southern blot or PCR analysis. *Cbl*$^{OST}$ allele: When tested on

BamHI-digested genomic DNA, the 5′ single-copy probe used to identify proper recombination events hybridized to a 16-kb wild-type fragment and to a 13.2-kb recombinant fragment. When tested on *BamHI*-digested genomic DNA, the 3′ single-copy probe used to identify proper recombination events hybridized to a 16-kb wild-type fragment and to a 7-kb recombinant fragment. *Cblb*$^{OST}$ allele: When tested on *Asp718*-digested genomic DNA, the 5′ single-copy probe used to identify proper recombination events hybridized to a 13.3-kb wild-type fragment and to a 4.7-kb recombinant fragment. When tested on *Asp718*-digested genomic DNA, the 3′ single-copy probe used to identify proper recombination events hybridized to a 13.3-kb wild-type fragment and to a 9.3-kb recombinant fragment. A neo$^r$-specific probe was used to ensure that adventitious non-homologous recombination events had not occurred in the selected clones corresponding to the *Cbl*$^{OST}$ and *Cblb*$^{OST}$ alleles.

### Production of knock-in mice

Mutant ES cells were injected into FVB blastocysts. In the case of the *Cblb*$^{OST}$ allele, excision of the frt-neo$^r$-frt cassette was achieved through cross with transgenic mice expressing a FLP recombinase under the control of the actin promoter (Rodriguez *et al*, 2000). In the case of the *Cbl*$^{OST}$ allele, screening for proper deletion of the loxP-tACE-CRE-PGK-gb2-neo-loxP cassette and for the presence of the sequence coding for the OST was performed by PCR using the following pair of primers: sense 5′-GTAGCTATCAACAAGGCGGAGGTGC-3′ and antisense 5′-AAAGGCAGGACCTTACTGTGACGTC-3′. This pair of primers amplified a 300-bp band in the case of the wild-type allele and a 457-bp band in the case of the *Cbl*$^{OST}$ allele. In the case of the *Cblb*$^{OST}$ allele, screening for proper deletion of the frt-neo$^r$-frt cassette and for the presence of the OST-(Stop)$_2$-IRES2-mTFP1-loxP sequence was performed by PCR using the following pair of primers: sense 5′-CTACTACATTCTCCCCCTAGATCCTAA-3′ and antisense 5′-GGCATGGACGAGCTGTACAAGTAAA-3′. This pair of primers amplified a 510-bp band in the case of the wild-type allele and a 202-bp band in the case of the *Cblb*$^{OST}$ allele. Note that the *Cblb*$^{OST}$ allele is a multitask allele permitting (i) affinity purification of the CBLB protein, (ii) constitutive or conditional deletion of the *Cblb* gene, and (iii) visualization of cells expressing the *Cblb* gene. The two last functionalities were not used in the present study.

### Mice

CBL$^{OST}$ (B6-*Cbl*$^{tm1Mal}$), CBLB$^{OST}$ (B6-*Cblb*$^{tm1Ciphe}$), *Cbl*$^{-/-}$ (B6.129-*Cbl*$^{tm1.1Hua}$), *Cblb*$^{-/-}$ (B6.129P2-*Cblb*$^{tm1Pngr}$), *Cd5*$^{-/-}$ (B6.129-*Cd5*$^{tm1Cgn}$), and wild-type B6 mice were maintained in specific pathogen-free conditions, and all experiments were done in accordance with institutional committees and French and European guidelines for animal care.

### Flow cytometry

Stained cells were analyzed using an LSRII system (BD Biosciences). Data were analyzed with the Diva software (BD Biosciences). Cell viability was evaluated using SYTOX Blue (Life Technologies). The following antibodies were used: anti-CD5 (53–7.3), anti-CD4 (RM4-5), anti-CD8α (53–6.7), and anti-CD45R (R3-6B2), all from BD Biosciences. For analyzing, the relative intracytoplasmic amounts of

CBL and CBLB in double-positive thymocytes and CD4[+] T cells from the spleen, cells were stained with antibodies directed at CD5 (53–7.3), CD8α (53–6.7), and CD4 (RM4-5), permeabilized with BD Cytofix/Cytoperm (BD Biosciences) for 30 min at 4°C, and then stained with saturating amount of Strep-Tactin APC (IBA GmbH).

## CD4[+] T-cell proliferation and IL-2 secretion

For proliferation and IL-2 secretion assay, purified CD4[+] T cells were stimulated with plate-bound anti-CD3 (145-2C11; Exbio) and soluble anti-CD28 (37–51; Exbio). After 48 h of culture, T-cell proliferation was assessed with CellTiter-Glo® Luminescent (Promega). The resulting luminescence, which is proportional to the ATP content of the culture, was measured with a Victor 2 luminometer (Wallac, Perkin Elmer Life Science). IL-2 production was measured with a DuoSet ELISA test (R&D Systems).

## CD4[+] T-cell isolation and short-term expansion

CD4[+] T cells were purified from pooled lymph nodes and spleens with a Dynabeads Untouched Mouse CD4[+] T Cell Kit (Life Technologies); cell purity was 95%. Purified CD4[+] T cells were expanded for 48 h with plate-bound anti-CD3 (145-2C11, 5 μg/ml) and soluble anti-CD28 (37–51; 1 μg/ml) both from Exbio. After 48 h of culture, CD4[+] T cells were harvested and grown in the presence of IL-2 (5–10 U/ml) for 48 h prior to AP-MS analysis.

## Stimulation and lysis of CD4[+] T cells prior to AP-MS analysis

Short-term expanded CD4[+] T cells ($100 \times 10^6$) from CBL[OST], CBLB[OST], and wild-type mice were left unstimulated or stimulated at 37°C with antibodies. In the last instance, CD4[+] T cells were incubated with anti-CD3 (0.2 μg per $10^6$ cells; 145-2C11; Exbio) and anti-CD4 (0.2 μg per $10^6$ cells; GK1.5; Exbio) on ice, followed by one round of washing at 4°C. Cells were then incubated at 37°C for 5 min and subsequently left unstimulated or stimulated at 37°C with a purified rabbit anti-rat (0.4 μg per $10^6$ cells; Jackson Immuno-Research) for 0.5, 2, 5, or 10 min at 37°C. Stimulation was stopped by the addition of a twice-concentrated lysis buffer (100 mM Tris, pH 7.5, 270 mM NaCl, 1 mM EDTA, 20% glycerol, 0.2% n-dodecyl-β-maltoside) supplemented with protease and phosphatase inhibitors. After 10 min of incubation on ice, cell lysates were centrifuged at 21,000 g for 5 min at 4°C. Postnuclear lysates were then used for affinity purification.

## Affinity purification of protein complexes

Equal amount of postnuclear lysates were incubated with prewashed Strep-Tactin Sepharose beads (IBA GmbH) for 1.5 h at 4°C on a rotary wheel. Beads were then washed five times with 1 ml of lysis buffer in the absence of detergent and of protease and phosphatase inhibitors. Proteins were eluted from the Strep-Tactin Sepharose beads with 2.5 mM D-biotin.

## Tandem MS analysis

Following affinity purification, protein samples were partially air-dried in a Speed-Vac concentrator, reconstituted in Laemmli buffer containing DTT (25 mM), and heated at 95°C for 5 min. Cysteines were alkylated for 30 min at room temperature by the addition of iodoacetamide (90 mM). Protein samples were loaded on an SDS–PAGE gel ($0.15 \times 3 \times 8$ cm) and subjected to electrophoresis. Migration was stopped as soon as the protein sample entered the gel. The gel was briefly stained with Coomassie blue, and a single slice containing the whole sample was excised. The gel slice was washed twice with 100 mM ammonium bicarbonate and once with 100 mM ammonium bicarbonate–acetonitrile (1:1). Proteins were in-gel-digested using 0.6 μg modified sequencing-grade trypsin (Promega) in 50 mM ammonium bicarbonate overnight at 37°C. The resulting peptides were extracted from the gel by one round of incubation (15 min, 37°C) in 50 mM ammonium bicarbonate and two rounds of incubation (15 min each, 37°C) in 10% formic acid–acetonitrile (1:1). The three extracted fractions were pooled with the solution in which the digestion proceeded and air-dried. Peptides were further purified on a C18 ZipTip (Millipore) and dried again. To identify CBL- and CBLB-binding partners, the tryptic peptides were resuspended in 20 μl of 2% acetonitrile and 0.05% trifluoroacetic acid and analyzed by MS. A mix of standard synthetic peptides (iRT Kit; Biognosys) was spiked in all samples to monitor the stability of the nano-LC-MS system during the analytical sequence. Peptides were analyzed by nano-liquid chromatography (LC) coupled to tandem MS using an UltiMate 3000 system (NCS-3500RS Nano/Cap System; Dionex) coupled to an Orbitrap Velos Pro mass spectrometer (Thermo Fisher Scientific). Five microliters of each sample was loaded on a C18 precolumn (300 μm inner diameter × 5 mm, Dionex) in a solvent made of 2% acetonitrile and 0.05% trifluoroacetic acid, at a flow rate of 20 μl/min. After 5 min of desalting, the precolumn was switched online with the analytical C18 column (75 μm inner diameter × 50 cm, in-house packed with Reprosil C18) equilibrated in 95% solvent A (5% acetonitrile, 0.2% formic acid) and 5% solvent B (80% acetonitrile, 0.2% formic acid). Peptides were eluted using a 5–50% gradient of solvent B over 105 min at a flow rate of 300 nl/min. The LTQ Orbitrap Velos was operated in data-dependent acquisition mode with Xcalibur software. Survey scan MS was acquired in the Orbitrap on the 350–2,000 $m/z$ range, with the resolution set to a value of 60,000. The 20 most intense ions survey scans were selected for fragmentation by collision-induced dissociation, and the resulting fragments were analyzed in the linear trap. Dynamic exclusion was used within 60 s to prevent repetitive selection of the same peptide. Technical LC-MS measurements were performed in triplicate (CBL-OST) and duplicate (CBLB-OST) for each of the samples.

## Protein identification and quantification

Raw MS files were processed with MaxQuant software (version 1.5.0) for database search with the Andromeda search engine and quantitative analysis. Data were searched against *Mus musculus* entries in the UniProt protein database (release UniProtKB/Swiss-Prot 2014_09; 16,699 entries). Carbamidomethylation of cysteines was set as a fixed modification, whereas oxidation of methionine, protein N-terminal acetylation, and phosphorylation of serine, threonine, and tyrosine were set as variable modifications. Specificity of trypsin digestion was set for cleavage after K or R, and two missed trypsin cleavage sites were allowed. The precursor

mass tolerance was set to 20 ppm for the first search and 4.5 ppm for the main Andromeda database search. The mass tolerance in tandem MS mode was set to 0.5 Da. Minimum peptide length was set to seven amino acids, and minimum number of unique peptides was set to one. Andromeda results were validated by the target decoy approach using a reverse database at both a peptide and protein false discovery rate of 1%. For label-free relative quantification of the samples, the match between runs option of MaxQuant was enabled with a time window of 0.5 min, to allow cross-assignment of MS features detected in the different runs.

### Identification of specific binding partners

To identify specific CBL- and CBLB-binding partners, we used the intensity metric from the MaxQuant protein group.txt output (sum of peptide intensity values for each protein) normalized by iRT standard peptides (Dataset EV1). Protein intensities were first normalized by adjusting the mean log intensities corresponding to all the proteins that were detected at all times points and in all biological and technical replicates of CBL$^{OST}$, CBLB$^{OST}$, and wild-type origins. To replace missing protein intensity values, we estimated the statistics of background signals from protein intensities of wild-type samples. Accordingly, for each detected protein, the mean and standard deviation of log-transformed wild-type intensities were calculated. Missing values were then replaced by sampling from a log normal distribution centered on the 5-percentile of the distribution of mean log wild-type intensities with a standard deviation set to the mean standard deviation of log wild-type intensities computed across all detected proteins. Mean log intensities (across all detected proteins) in wild-type and CBL$^{OST}$ and CBLB$^{OST}$ backgrounds were then normalized independently for each stimulatory condition.

To determine whether a given detected protein was specifically associated with the bait at a time "$t$", we compared the distributions of these normalized log intensities between wild-type and CBL$^{OST}$ (or CBLB$^{OST}$) backgrounds by computing the $P$-value from a one-way ANOVA test "$P(t)$" along with the corresponding enrichment "$r(t)$". Proteins were selected as specific partners of the CBL$^{OST}$ (or CBLB$^{OST}$) bait when both the $P$-value was below a set threshold ($P = 0.001$ for CBL$^{OST}$ and $P = 0.005$ for CBLB$^{OST}$) and the corresponding enrichment was greater than twofold. This selection process was repeated 200 times using wild-type and CBL$^{OST}$ (or CBLB$^{OST}$) intensities resampled independently from the original distributions using a bootstrap algorithm. Only proteins that were selected as specific partners in at least 90% of such tests were included in the final list of specific partners.

### Construction of correlation networks

To quantify the kinetics of recruitment of a given interactor to the CBL-OST (or CBLB-OST) bait, we first normalized protein intensities for each condition to that of the corresponding bait. This ensured a constant intensity for the CBL-OST (or CBLB-OST) bait across all conditions (all times points and all biological and technical replicates). Next, for each protein and each biological replicate, the intensity was normalized to the maximum (across all time points) of the mean intensities computed across technical replicates. From these normalized recruitment intensities, we quantified the correlation in recruitment between two different proteins by computing the

Pearson coefficient along with the corresponding $P$-value. To avoid the detection of spurious correlations, we used a bootstrap algorithm to resample the normalized recruitment intensities from their original distributions. The Pearson coefficient and the corresponding $P$-value were then estimated as their mean values across 1,000 bootstrap samples. This procedure was conducted for all pairs of identified proteins to construct a correlation matrix ($R_{ij}$) along with the corresponding matrix of $P$-values ($P_{ij}$). To obtain the correlation network for a chosen significance level $\alpha$ (we arbitrarily chose $\alpha = 5 \times 10^{-5}$ here) between the $N$ identified interactors (excluding the bait), we applied Bonferroni correction for $N(N-1)/2$ tests yielding a threshold $P$-value $*P = 5.29 \times 10^{-8}$ for the CBLB correlation network and $*P = 1.05 \times 10^{-8}$ for the CBL correlation network.

### Comparison with interactions reported in the BioGRID database

Physical interactions reported in version 3.4.130 of the BioGRID database among proteins from *Homo sapiens* or *Mus musculus* were used in our analysis.

### Immunoprecipitation, Western blot analysis, and antibodies

Briefly, expanded CD4$^+$ T cells of the specified genotypes were incubated with anti-CD3 (0.2 μg per $10^6$ cells; 145-2C11; Exbio) and anti-CD4 (0.2 μg per $10^6$ cells; GK1.5; Exbio) for 15 min on ice, followed by one round of washing at 4°C. Cells were then incubated at 37°C for 5 min and subsequently left unstimulated or stimulated at 37°C by cross-linking for 1 min with purified goat anti-rat IgG F(ab′)$_2$ (0.4 μg per $10^6$ cells; Jackson Immuno-Research). Stimulation was stopped by the addition of a twice-concentrated lysis buffer (100 mM Tris, pH 7.5, 270 mM NaCl, 1 mM EDTA, 20% glycerol, 0.2% n-dodecyl-β-maltoside) supplemented with protease and phosphatase inhibitors. After 10 min of incubation on ice, cell lysates were centrifuged at 21,000 $g$ for 5 min at 4°C. Postnuclear lysates were used for immunoprecipitation or as whole-cell lysates for subsequent immunoblot analysis. For immunoprecipitation, equal amounts of cell lysates were incubated for 1.5 h with specified antibodies. Immune complexes were purified with Pansorbin (Calbiochem) and were washed three times before elution in SDS-containing sample buffer. Eluted samples and whole-cell lysates were loaded on 8% SDS–PAGE gel and subsequently analyzed by immunoblot with specific antibodies. The following antibodies were used for immunoprecipitation: anti-CBL (sc-170), anti-CBLB (sc-1704), anti-CSK (sc-286), and anti-CRKL (sc-319) from Santa Cruz Biotechnology, and anti-p85-PI3K (4292) from Cell Signaling Technology. The following antibodies were used for immunoblot analysis: anti-CBL (2747), anti-p85-PI3K (4292), and anti-PI3K p110α (4249) from Cell Signaling Technology; and anti-CD5 (sc-6986), anti-CSK (sc-166513), anti-CRKL (sc-365471), and anti-CBLB (sc-376409) from Santa Cruz Biotechnology. The antibody anti-phospho-LCK(Y505) (2751) from Cell Signaling Technology was used for intracellular staining.

### Data availability

Dataset EV2 shows the quantification of all peptides identified in this study. Selected partners and corresponding peptides of CBL and

CBLB are also listed. The mass spectrometry proteomics data have been deposited to the ProteomeXchange Consortium via the PRIDE partner repository (http://www.ebi.ac.uk/pride) with the dataset identifier PXD004130.

**Expanded View** for this article is available online.

## Acknowledgements

We thank H. Gu and J. Penninger for CBL- and CBLB-deficient mice, F. Lozano and Marc Orta for CD5-deficient mice, and Ingrid Hinchliffe and Fabien Danjan for the construction of CBL^OST and CBLB^OST mice. This work was supported by CNRS, INSERM, European Research Council (FP7/2007–2013 grant no. 322465 ("Integrate") to B.M), Agence Nationale de la Recherche (PHENOMIN project to B.M. and Basilic project to M.M.), the Investissement d'Avenir program ProFI (Proteomics French Infrastructure project; ANR-10-INBS-08) and PHENOMIN (French National Infrastructure for Mouse Phenogenomics), Région Midi-Pyrénées, and by fellowships from the Integrate project (G.V. and E.B.), the NANOASIT Euronanomed project (A.G.B.), and the PHENOMIN project (L.G.).

## Author contributions

BM, RR, and GV conceived the project; GV and RR designed and did the experiments for Figs 1, 4–7 and EV5; AGdP, AG-B, RR, KC, and OB-S designed and did the experiments for Fig 2 and Dataset EV1; GV designed and performed the computational and bioinformatics analysis for Figs 2–6 and EV2–EV4; BM, FF, and MM designed and performed the experiments for Fig EV1 with the technical help of LG and EB; and GV, RR, and BM wrote the manuscript.

## Conflict of interest

The authors declare that they have no conflict of interest.

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
