## [Review Process File · Molecular Systems Biology]

Co-recruitment analysis of the CBL and CBLB signalosomes in primary T cells identifies CD5 as a key regulator of TCR-induced ubiquitylation

Guillaume Voisinne, Antonio García Blesa, Karima Chaoui, Frédéric Fiore, Elise Bergot, Laura GIRARD, Marie Malissen, Odile Bulet-Schiltz, Anne Gonzalez-de-Peredo, Bernard Malissen and Romain Roncagalli

Corresponding authors: Romain Roncagalli and Bernard Malissen, CNRS

Review timeline:	Submission date:	26 January 2016
	Editorial Decision:	04 March 2016
	Revision received:	11 May 2016
	Editorial Decision:	08 June 2016
	Revision received:	14 June 2016
	Accepted:	23 June 2016

Editor: Thomas Lemberger

Transaction Report:

1st Editorial Decision

04 March 2016

Thank you again for submitting your work to Molecular Systems Biology. We have now heard back from the four referees who agreed to evaluate your manuscript. As you will see from the reports below, the referees find the topic of your study of potential interest. They raise, however, substantial concerns on your work, which should be convincingly addressed in a major revision of this work.

The major points raised by the reviewers refer to the following issues:

- while the reviewers find the correlation network approach interesting, they feel the further systematic validation of a sufficient subset of interactors would be useful to benchmark better the method and bolster the value of the dataset as a resource.
- some of the functional implications of CD5 as a key regulator should be examined in more depth; while a full mechanistic investigation of all the aspects noted by reviewer #2 and #4 would be beyond the scope of this study, demonstrating the functional relevance (eg on LCK activity) of CD5:CBL-mediated CSK recruitment would be important.
- the absence of interaction LAT with Cbl should be clarified in view of previous literature, including one of your own studies (Roncagalli et al 2014).

- the raw mass spectrometry data should be deposited in a public database as well as the validated interactions. Please include the respective identifiers/accession numbers/links in a separate Data availability sections at the end of the Materials & Methods section.

REFeree COMMENTS

Reviewer #1:

In this manuscript, Voisinne et al. performed a time-resolved quantitative proteomic analysis of CD4 T cells to compare the interactome of Cbl and Cbl-b in response to TCR stimulation. In their system, T cells were isolated from mice that expressed a genetic tag at the carboxyl-terminus of Cbl and Cbl-b proteins, allowing for the immunoprecipitation of these proteins using a streptavidin derivative that binds with high affinity to the genetic tag sequence. Analysis of the MS data provided a comprehensive representation of the Cbl and Cbl-b interactomes across a time course of TCR stimulation, highlighting the both the redundant and unique functional features of these proteins. Furthermore, the authors generated co-recruitment correlation networks from the MS data as an approach to identify the underlying protein-protein interactions responsible for the assembly of Cbl and Cbl-b interactomes.

Overall, data presented in this manuscript will be a useful resource for research groups interested in these ubiquitin ligases and/or the identified protein interactors.

Major concerns:

1. Raw MS data files and search results should be uploaded on a publically available database to allow reviewers to look over the data. Notably, in the given supplemental data, it is unclear which peptides were identified from the MS runs, and to which proteins the peptides corresponded.
2. The authors should analyze the reproducibility of the MS runs across technical and biological replicates. For example, the log(peak areas) for each identified peptide in replicate 1 vs. replicate 2 etc. should be plotted to give the readers a sense of the quality of the MS data. Furthermore, the peptide peak intensities of Cbl/Cbl-b should be illustrated across sample runs to assess the reproducibility of the IP, sample prep, and MS acquisition.
3. The fundamental assumption made by the authors in generating the co-recruitment correlation network (CN) is that if two recruited proteins are interacting with each other, their kinetics of recruitment over the time course should be similar. The authors go on to compare their predicted CN networks to Cbl and Cbl-b interacting partners reported in the BioGRID database. To strengthen the computational model used by the authors to generate the CN networks, validation of a substantial number of interactors should be performed using immunoblots.
4. Some of readers of this work, especially those with a background in immunology, will likely not have a strong background in biostatistics/bioinformatics and as such it would be beneficial to explain the approaches used to generate the CN networks in way that is less technical.
5. In Figure 1A, pull down experiments reveal equivalent levels of Cbl-b protein expression; however, in the whole cell lysates, Cbl-b expression drastically decreases with TCR stimulation. Can the authors comment on these observations?
6. Does the epitope tag added alter the DUB domain function of either tagged protein. At least some control should be done to assess this.

Minor concerns:

1. Stimulation of CD4 T cells prior to AP-MS analysis was performed with anti-CD3 and anti-CD4 incubation at 4C followed by crosslinking at 37C. Various publications show that temperature shifts can activate the TCR signaling pathway (Mahammad et al. *Biochim Biophys Acta*, 2010, doi: 10.1016/j.bbali.2010.02.003; Magee et al. *J Cell Sci*, 2005, doi: 10.1242/jcs.02442; Ji et al., *J Prot Res*, 2015 doi: 10.1021/pr501172u).
2. In the methods section, should include the concentration of plate-bound CD3 and soluble CD28 used for proliferation and expansion.
3. For each Cbl/Cbl-b interactor, the number of peptides identified unique to the protein should be represented in the supplement.

Reviewer #2:

This paper presents an excellent analysis of the composition and assembly kinetics of Cbl and Cbl-b interactomes after TCR stimulation of mature CD4⁺ T cells. The authors identified multiple novel Cbl and Cbl-b binding partners (98 for Cbl and 43 for Cbl-b), and analysed the timing of recruitment of Cbl/ Cbl-b binding partners in the Cbl/Cbl-b interactomes in order to computationally identify interactions between these proteins. The predictive power of this approach to identify protein-protein interactions was then validated by checking the newly postulated interactions against a database of known protein interactions. This computational approach identified Csk and CrkL as novel CD5 binding partners, and these new interactions have been experimentally validated using immunoprecipitation. Additionally, this analysis showed that recruitment of CD5 to Cbl/Cbl-b was correlated with that of ubiquitin, leading the authors to suggest that CD5 regulates Cbl/Cbl-b access to and ubiquitination of membrane-proximal TCR signalosome proteins. This hypothesis was partially experimentally verified by demonstrating that absence of CD5 reduced ubiquitinated proteins immunoprecipitated with Cbl or Cbl-b. These findings have important implications for better understanding of negative regulation of T cell receptor signalling networks.

However, the significance of the major novel findings of this paper (the newly identified Cbl and Cbl-b binding partners, CD5 interactions with Csk and CrkL, the role of CD5 in regulation of Cbl and Cbl-b interactions with ubiquitinated proteins) is unclear. The authors discuss some interesting potential implications of the new findings (suggested role for CD5 in recruitment of Cbl and Cbl-b to cell membrane, leading to regulation of Lck phosphorylation through CD5-Cbl complex-mediated recruitment of Csk to cell membrane), but there are no experimental data to support these hypotheses. This is a major limitation that severely reduces the significance of the paper.

Minor points:

- 1) The discussion is very interesting, but some parts read like a catalogue of interacting proteins. More focus on the novel/critical binding partners could make it easier for the reader.
- 2) The authors discuss the possible implications of the finding that Cbl-b, but not Cbl, interacts with LAT. However, a previous paper from the Malissen laboratory (Roncagalli et al. (2014) Quantitative proteomic analysis of signalosome dynamics in primary T cells identifies the CD6 surface receptor as a Lat-independent TCR signaling hub), using a similar strategy to characterize the LAT interactome, reported both Cbl and Cbl-b interactions with LAT. Is this difference the result of the current use of more stringent data analysis criteria?
- 3) On page 17:
However, the lack of phenotype of PAG-deficient mouse together with recent data suggest that additional proteins can compensate for the loss of PAG and contribute to recruit CSK at the inner face of the plasma membrane following TCR stimulation (Dobenecker et al, 2005; Reginald et al, 2015; Xu et al, 2005). Our AP-MS analysis identifies CSK as one of those candidate proteins.

Shouldn't the last sentence be: Our AP-MS analysis identifies CD5 as one of those candidate proteins?

4) What is the time of TCR stimulation in Figure 7?

Reviewer #3:

Comments to the authors:

The CNRS lab from Romain Roncagalli and Bernhard Malissen is at the forefront of TCR signaling network analyses in a systems biology approach. They now have been generated GM mice to express a tagged form of CBL or CBLB, respectively that allowed precipitation by Strep-Tactin-coated beads to identify CBL/CBLB interacting protein candidates as well as determining stimulation-induced dynamic changes in these signalling complexes over time under defined stimulation strength.

Although the basic regulatory steps have been elucidated before, many features of the CBL/CBLB pathway(s) is/are only begin to emerge. Description of the kinetic behaviour and the novel CD5 scaffold function of these E3 ligases thus have made the next step in resolving this rather complex story.

This study is technically well performed and the results reported do fully support the main conclusions. The data are novel and will be of interest for immunologists working on molecular mechanisms of T cell activation.

Taken together, their work represent a very careful study and i.e. based on their in part distinct PPIs, conveys a potentially important message for a subset selectivity of CBL/CBLB pathway(s) functions.

No revision needed.

Reviewer #4:

Summary:

In the paper by Voisinne et al., the authors create mice genetically engineered to express the one-STrEP epitope tag on endogenous Cbl or Cblb. They demonstrate that the T cells from these mice express the epitope tagged Cbl or Cblb at the same levels as Cbl or Cblb in wild type mice and they show that the number and function of T cells is normal. They then use peripheral T cells from these mice to pull down either Cbl or Cblb to assess the interacting partners with these proteins over a time course of T cell receptor activation. The work builds an interacting network of proteins that includes both unique and overlapping proteins within the Cbl and Cblb interaction networks. Their data also have identified numerous interacting proteins not previously described in the BioGrid database. They validate the interactions of CSK, CRKL, and CD5 and Cblb by co-immunoprecipitation experiments. They validate the interactions between CD5 and Cbl by co-immunoprecipitation experiments. They show that upon TCR activation, Cblb co-immunoprecipitates more ubiquitinated proteins than Cbl but interestingly, that when Cbl is precipitated from Cblb null mice, more ubiquitinated proteins co-precipitate with Cbl. Finally they demonstrate that there is less ubiquitinated proteins associated with Cbl or Cblb when they are precipitated from mice that are CD5 null. They conclude that CD5 regulates TCR stimulated Cbl and Cblb mediated ubiquitination.

General remarks:

Cbl and Cblb have been shown to play roles as positive and negative regulators of many signaling systems such as receptor tyrosine kinases and the T cell receptor. Previous work has suggested that the two proteins have differential effects on TCR signaling, in particular Cblb is a potent negative regulator of the costimulatory pathways while Cbl is not. Overall the paper provides a catalogue interacting proteins that associate with Cbl and Cblb upon TCR activation (the majority of the interactions described were increased upon activation) and provide network analyses that allow the reader to see overlap, similarities, and differences in the spectrum of interacting proteins between

Cbl and Cblb. There are many previously undescribed interactions and the direct comparison of Cbl and Cblb interactors adds value to this work. This certainly be of interest to investigators studying T cell receptor activation and function and to a lesser extent it will be of interest to investigators studying other systems where Cbl proteins play a role.

Major points:

1. The title of the paper focuses on the role of CD5 as a key regulator of Cbl mediated ubiquitination. CD5 has previously been shown to negatively regulate T cell receptor signaling and further recent papers have shown that Cbl ubiquitinates and down regulates CD5 (Demydenko, BBRC, 392: 400-5, 2010) and that Cbl binds to the cytoplasmic tail of CD5 and that this binding was necessary for the down regulation of VAV, one of the Cbl substrates that is down regulated upon TCR activation (e.g. Roa et al, BBRC, 432:52, 2013). Thus the work showing that both Cbl and Cblb associate with CD5 and that the amount of ubiquitinated proteins that coprecipitate with either Cbl protein is decreased in the absence of CD5 is of interest. However, the work is well short of demonstrating the basis for the increase in the co-precipitation of ubiquitinated proteins with Cbl in the absence of Cblb and elucidating the functional role of CD5 in the observed changes in ubiquitination. Overall the authors need to address more rigorously the changes in ubiquitination that they describe. For example, is the increase in the co-precipitated ubiquitinated proteins seen with Cbl when Cblb is absent due to competition between Cbl and Cblb so that more Cbl associates with CD5 when Cblb is absent (and thus that Cblb is the primary protein responsible for CD5 ubiquitination in wt cells)? The data in Figure 6C suggests more association between Cbl and CD5 when Cblb is absent. What happens to TCR and CD5 ubiquitination and downregulation and signaling when either Cblb or Cbl is lost. Also what happens to TCR ubiquitination and downregulation and signaling when CD5 is lost?

2. The loss of Cbl has minimal effects on TCR signaling in peripheral T cells but Cblb has been implicated to regulate the costimulatory pathway in T cells upon TCR activation. What happens to these pathways when they activate the cells in the absence of CD5?

Minor points:

1. A significant body of data has demonstrated binding of Cbl to LAT and regulation of LAT signaling clusters by Cbl, including downregulation of the TCR and LAT by Cbl. However, the mass spec analysis did not identify LAT in the Cbl interacting networks and the authors make the point that they found Cblb but not Cbl. This needs to be discussed in . Is it due to the use of peripheral T cells vs thymocytes or is a technical limitation of the study?

2. The authors do reference the Roa paper cited above as evidence that Cbl associates with CD5 but they do not discuss the fact that the work in that paper also implicates the interaction in down regulation of VAV upon TCR activation. They should include this in the discussion.

3. Page 11, line 22: in the phrase "Interestingly, in some species, ubiquitin remains fused to S27A following its incorporation in the mature ribosome", would be clearer as "Interestingly, in some species, ubiquitin remains fused to S27A that is incorporated into the mature ribosome.

4. Page 12, line 7, : the authors should use "large number of..." instead of "large numbers of..."

5. Page 13, line 17: "To further refined" should be replaced with "To further refine".

6. Page 21, line 7: in the sentence "Our AP-MS analysis identifies CSK as one of those candidate proteins", the authors probably meant CD5 instead of CSK.

7. Page 49: the legend for Figure 6C has to be switched with the legend for Figure 6D.

(see next page)

Responses to the Editor and the Reviewers are shown in bold.

Dear Dr Malissen,

Thank you again for submitting your work to Molecular Systems Biology. We have now heard back from the four referees who agreed to evaluate your manuscript. As you will see from the reports below, the referees find the topic of your study of potential interest. They raise, however, substantial concerns on your work, which should be convincingly addressed in a major revision of this work.

The major points raised by the reviewers refer to the following issues:

- while the reviewers find the correlation network approach interesting, they feel the further systematical validation of a sufficient subset of interactors would be useful to benchmark better the method and bolster the value of the dataset as a resource.

To further highlight the predictive power of our correlation network approach, an additional set of interactions predicted by our computational approach was subjected to experimental validation (that is coimmunoprecipitation studies). It corresponds to the high confidence CRKL and CSK first neighbors subnetworks as shown in our original Figure 5 C. Based on commercially available specific antibodies, we were able to test 11 out of the 19 predicted interactions found in this subnetwork. We successfully validated by co-immunoprecipitation 9 of the 11 tractable interactions. Importantly, 8 out of the 9 interactions that we have validated were not previously reported in BioGRID (see Figure 5-C). Therefore, our novel data benchmark the method described in our manuscript. They have been added in the novel Figure 5 (panels D to H) and discussed on pages 15 and 16 of the revised manuscript.

- some of the functional implications of CD5 as a key regulator should be examined in more depth; while a full mechanistic investigation of all the aspects noted by reviewer #2 and #4

would be beyond the scope of this study, demonstrating the functional relevance (eg on LCK activity) of CD5:CBL-mediated CSK recruitment would be important.

We agree that a full mechanistic investigation of all the aspects noted by reviewer #2 and #4 is beyond the scope of this study and as suggested focused on the functional relevance of the novel interaction between CD5 and CSK reported in our manuscript. To test whether CD5 regulates the activity of CSK following TCR activation, we compared the phosphorylation of the negative-regulatory tyrosine found at position 505 of LCK, since it constitutes an excellent read-out of CSK kinase activity. To assess the role of CD5 on the negative regulation of LCK via CSK-mediated phosphorylation of Y505, we followed a conditional density-based analysis of T cell signaling in single-cell as recently described in two publications in Science and PNAS (doi: 10.1073/pnas.1419337111 and doi: 10.1126/science.1250689.). Accordingly, we used the intrinsic variability in CD5 expression that exists among CD4⁺ T cells and can be measured by flow cytometry (see panel A in Figure R that is provided for Reviewers #2 and #4). It allowed us to define different populations of CD4⁺ T cells based on their expression of CD5. (CD5^{lo}, CD5^{med}, and CD5^{high}). By performing intracellular staining of those cells using an antibody specific for phospho-LCKY505, we observed a continuous increase in the phosphorylation of the LCKY505 as a function of CD5 expression (Figure R2B, C). Therefore such

Figure R2. CD5 contributes to control the phosphorylation of the negative-regulatory tyrosine found at position 505 of LCK via CSK.

Analysis by flow cytometry of the expression of CD5 and of the phosphorylation of tyrosine 505 of LCK. CD4⁺ T cells were stimulated by cross-linking CD3 and CD4 for 1min. Stimulated cells were fixed with 1.6% PFA at room temperature for 15 min and permeabilized in 90% MetOH on ice for 10 min. Cells were then stained with the indicated antibodies and fluorescence was acquired on a LSR II flow cytometer.

A The depicted gates define three populations of T cells with different expression of CD5 at their surface (CD5^{lo}, CD5^{med} and CD5^{hi}).

B The histogram represents the levels of phospho-LCKY505 found in the three populations defined in A.

C The upper panel represents the mean (+/- SD) of the log fluorescence intensity of phospho-LCKY505 as a function of the geometric mean fluorescence intensity of CD5. Mean and SD were computed from populations of cells using the regular binning of the CD5 expression histogram (lower panel).

D Effect of CD5 cross-linking on the phosphorylation of Y505 of LCK. CD4⁺ T cells were stimulated with anti-CD3 plus anti-CD4 in the presence or absence of anti-CD5. The intensity of phospho-LCKY505 is represented as percent of phospho-LCKY505 intensity in unstimulated condition.

results suggest that CD5 contributes to control the phosphorylation of the negative-regulatory tyrosine found at position 505 of LCK via CSK.

To further test that CD5 directly regulates CSK activity, we stimulated CD4⁺ T cells with anti-CD3 plus anti-CD4 in the presence or absence of anti-CD5 (Figure R2D). Cross-linking TCR-CD4 molecules with CD5 increased the phosphorylation of Y505 of LCK. Therefore, CD5 contributes to control the phosphorylation of the negative-regulatory tyrosine found at position 505 of LCK via CSK. Altogether, these results suggest the functional relevance of the novel CD5-CSK interaction reported in our manuscript. Please let us know if you want us to show those data as an additional Figure EV or keep them in the present point-by-point responses that intend to be also published.

- the absence of interaction LAT with Cbl should be clarified in view of previous literature, including one of your own studies (Roncagalli et al 2014).

We fully agree with the reviewers and thus clarified this issue below:

- 1. Phenotypic and functional relationships between LAT and CBL have been clearly demonstrated in the literature. Several studies highlighted the importance of CBL for the internalization and ubiquitylation of LAT (see Mol. Cell. Biol., 2007; 27(24): 8622-8636 and PNAS, 2011; 108(7): 2885-2890.), but none of them concluded that a potential (direct or indirect) physical interaction exists between LAT and CBL. In the BioGRID database, a physical interaction between LAT and CBL is reported but it is based on a unique study (Jiang et al., Leukemia Research, 2007) where the identification of CBL solely relied on its molecular weight (!);the corresponding band being NOT probed with an anti-CBL antibody.**
- 2. As appropriately mentioned by the reviewers, we recently published a study showing that the adaptor LAT could associate with the ubiquitin ligases CBL and CBLB (Roncagalli et al 2014). We would like to draw the attention of the reviewers to some technical aspects that distinguish our former study from the current one. Instead of using anti-CD3 and anti-CD4 antibodies to stimulate T**

cells as in the present study, our previous study was based on pervanadate, a potent phosphatase inhibitor that is used as a surrogate for TCR stimulation. It is possible that pervanadate triggered a more potent and broader stimulation (in terms of the number of phosphorylation sites) than the more physiological mode of stimulation used in the present study. Hence, pervanadate stimulation could induce a greater number of protein-protein interactions potentially including some adventitious ones. In addition, the number of spectral counts reported in our LAT interactome (Roncagalli et al 2014) was much higher for peptides corresponding to CBLB than for those corresponding to CBL (Nat Immunol. 2014 Apr;15(4):384-92; Supplementary Table 1). In line with the results reported in our present study, these last observations suggest that LAT preferentially associates with CBLB rather than with CBL.

- the raw mass spectrometry data should be deposited in a public database as well as the validated interactions. Please include the respective identifiers/accession numbers/links in a separate Data availability sections at the end of the Materials & Methods section.

As indicated in the Data availability section, The mass spectrometry proteomics data have been deposited to the ProteomeXchange Consortium via the PRIDE partner repository (<http://www.ebi.ac.uk/pride>) with the dataset identifier PXD004130.

- we would also encourage you to include the source data for figure panels that show essential data, so that readers can download these data directly from the figure. Source data files are associated to individual panels of main figures. *Numerical data* should be provided as individual .xls files (including a tab describing the data) or csv or tab-delimited text files. *For 'blots' or microscopy*, uncropped images should be submitted. For *network visualization*, Cytoscape session files, if available, can be supplied. The files should be labeled as "Source Data for Figure 1A" etc. Source Data for Expanded View and Appendix figures should be uploaded as a single ZIP file containing all the Source Data for Expanded View and Appendix content. (Additional information on source data is available in the "Guide for Authors" section at <http://msb.embopress.org/authorguide#sourcedata>).

To comply with the above suggestions, we included a novel Table that corresponds to the source data for Figure 2 and in which we quantified all peptides identified in this study. Selected partners and corresponding peptides of CBL and CBLB were also listed.

Reviewer #1:

In this manuscript, Voisinne et al. performed a time-resolved quantitative proteomic analysis of CD4 T cells to compare the interactome of Cbl and Cbl-b in response to TCR stimulation. In their system, T cells were isolated from mice that expressed a genetic tag at the carboxyl-terminus of Cbl and Cbl-b proteins, allowing for the immunoprecipitation of these proteins using a streptavidin derivative that binds with high affinity to the genetic tag sequence. Analysis of the MS data provided a comprehensive representation of the Cbl and Cbl-b interactomes across a time course of TCR stimulation, highlighting the both the redundant and unique functional features of these proteins. Furthermore, the authors generated co-recruitment correlation networks from the MS data as an approach to identify the underlying protein-protein interactions responsible for the assembly of Cbl and Cbl-b interactomes.

Overall, data presented in this manuscript will be a useful resource for research groups interested in these ubiquitin ligases and/or the identified protein interactors.

Major concerns:

1. Raw MS data files and search results should be uploaded on a publically available database to allow reviewers to look over the data. Notably, in the given supplemental data, it is unclear which peptides were identified from the MS runs, and to which proteins the peptides corresponded.

The mass spectrometry proteomics data have been deposited to the ProteomeXchange Consortium via the PRIDE partner repository (<http://www.ebi.ac.uk/pride>) with the dataset identifier PXD004130. Moreover, we have added a novel Excel Table corresponding to the source data for Figure 2 and in which we quantified all peptides identified in this study. Selected partners and corresponding peptides of CBL and CBLB were also listed.

2. The authors should analyze the reproducibility of the MS runs across technical and biological replicates. For example, the log(peak areas) for each identified peptide in replicate 1 vs. replicate 2 etc. should be plotted to give the readers a sense of the quality of the MS data. Furthermore, the peptide peak intensities of Cbl/Cbl-b should be illustrated across sample runs to assess the reproducibility of the IP, sample prep, and MS acquisition.

To address Reviewer #1 concern, we have included in the revised version of our manuscript (see in the Extended View) a new Figure illustrating the reproducibility of the MS runs across technical and biological replicates (Figure EV2A, B). This novel figure also displays the intensities of peptides corresponding to CBL and CBLB across all sample runs (Figure EV2C, D). A sentence referring to this novel figure was also included on page 8 of the main text: “*The reproducibility of the AP-MS process was assessed for each condition of stimulation across technical and biological replicates (Fig. EV2).*”

3. The fundamental assumption made by the authors in generating the co-recruitment correlation network (CN) is that if two recruited proteins are interacting with each other, their kinetics of recruitment over the time course should be similar. The authors go on to compare their predicted CN networks to Cbl and Cbl-b interacting partners reported in the BioGRID database. To strengthen the computational model used by the authors to generate the CN networks, validation of a substantial number of interactors should be performed using immunoblots.

We thank Reviewer #1 for raising this important issue. We have carefully addressed it on pages 1 and 2 of the present point-by-point responses.

4. Some of readers of this work, especially those with a background in immunology, will likely not have a strong background in biostatistics/bioinformatics and as such it would be beneficial to explain the approaches used to generate the CN networks in way that is less technical.

We agree with Reviewer #1. Accordingly, we modified the main text (Page 13) to explain the approach used to generate the CN in more general terms and we orientate readers to the Materials and Methods section for additional technical details.

5. In Figure 1A, pull down experiments reveal equivalent levels of Cbl-b protein expression; however, in the whole cell lysates, Cbl-b expression drastically decreases with TCR stimulation. Can the authors comment on these observations?

We fully agree with the reviewer and we also spotted this point. In the original manuscript we intentionally showed a long exposure of the CBLB pulldown to show that ubiquitylated forms of CBLB occurred when the TCR is engaged (seen in both Fig 6A and Fig 1A). In the original manuscript we commented Fig 6A as follow: “TCR stimulation resulted in some CBLB degradation and in the detection of a higher molecular weight CBLB form, likely due to its ubiquitylation (marked with an asterisk). In contrast, no

degradation of CBL was detectable upon TCR stimulation and its molecular weight remained unchanged.” To clarify this issue, we have provided in the revised version of Fig 6A a short exposure of the Western blot corresponding to the same experiment. It clearly shows that the levels of CBLB degradation occurring in the CBLB pull down species are similar to those observed in whole cell lysates.

6. Does the epitope tag added alter the DUB domain function of either tagged protein. At least some control should be done to assess this.

CBL and CBLB are E3 ubiquitin-protein ligases and do not contain a DUB domain. DUB domain are found in deubiquitinating enzymes (DUBs) that are a large group of proteases that cleave ubiquitin from proteins and other molecules (for a recent review see: doi: 10.12688/f1000research.7220.1). CBL and CBLB do contain a ubiquitin association (UBA) domain a their carboxy-terminus (see <http://www.uniprot.org/uniprot/P22682> and Q3TTA7). The UBA of CBLB has a higher affinity for ubiquitin than that of CBL (Davies et al 2004; Zhou et al. 2008). Consistent with that view ubiquitin was 4.7 times more enriched in the CBLB interactome than in that of CBL (see main text pages 10 and 11). Therefore together with our extensive functional data demonstrating that the OST do not affect in a measurable manner the development and function of T cells, the above data suggests that the introduction of a OST at the carboxy-terminus of CBL and CBLB does not affect the function of the UBA domain.

Minor concerns:

1. Stimulation of CD4 T cells prior to AP-MS analysis was performed with anti-CD3 and anti-CD4 incubation at 4C followed by crosslinking at 37C. Various publications show that temperature shifts can activate the TCR signaling pathway (Mahammad et al. Biochim Biophys Acta, 2010, doi: 10.1016/j.bbaliip.2010.02.003; Magee et al. J Cell Sci, 2005, doi: 10.1242/jcs.02442; Ji et al., J Prot Res, 2015 doi: 10.1021/pr501172u).

We would like to thank Reviewer #1 for raising this extremely important technical aspect. We were aware of the provided publications and we also observed temperature shifts effects on global phosphotyrosine patterns. To avoid caveats resulting from such effects, we incubated our T cells at 37°C for 5 minutes before stimulation. We and others (Journal of Cell Science, 2005, 118, 3141-3151) have observed that a 5 minutes incubation at 37°C is sufficient to completely reverse the phosphotyrosine increase in tyrosine phosphorylation induced by low temperature. This key point was mistakenly omitted in the original description of the protocol corresponding to the MS experiments. We have thus corrected this section in the revised manuscript (page 27) by adding that the T cells were additionally incubated at 37°C for 5 min.

2. In the methods section, should include the concentration of plate-bound CD3 and soluble CD28 used for proliferation and expansion.

We thank Reviewer #2 for raising this issue. The Methods section has been modified accordingly (page 23).

3. For each Cbl/Cbl-b interactor, the number of peptides identified unique to the protein should be represented in the supplement.

We would like to draw attention of Reviewer #2 to the fact that this particular information is already available in the Dataset EV1, sheet A and B, columns “Peptide counts (unique)”. These unique peptides are also listed in a novel Excel Table that corresponds to the source data for Figure 3 in which we quantified all peptides identified in this study.

Reviewer #2:

This paper presents an excellent analysis of the composition and assembly kinetics of Cbl and Cbl-b interactomes after TCR stimulation of mature CD4+ T cells. The authors identified

multiple novel Cbl and Cbl-b binding partners (98 for Cbl and 43 for Cbl-b), and analysed the timing of recruitment of Cbl/ Cbl-b binding partners in the Cbl/Cbl-b interactomes in order to computationally identify interactions between these proteins. The predictive power of this approach to identify protein-protein interactions was then validated by checking the newly postulated interactions against a database of known protein interactions. This computational approach identified Csk and CrkL as novel CD5 binding partners, and these new interactions have been experimentally validated using immunoprecipitation. Additionally, this analysis showed that recruitment of CD5 to Cbl/Cbl-b was correlated with that of ubiquitin, leading the authors to suggest that CD5 regulates Cbl/Cbl-b access to and ubiquitination of membrane-proximal TCR signalosome proteins. This hypothesis was partially experimentally verified by demonstrating that absence of CD5 reduced ubiquitinated proteins immunoprecipitated with Cbl or Cbl-b. These findings have important implications for better understanding of negative regulation of T cell receptor signalling networks.

However, the significance of the major novel findings of this paper (the newly identified Cbl and Cbl-b binding partners, CD5 interactions with Csk and CrkL, the role of CD5 in regulation of Cbl and Cbl-b interactions with ubiquitinated proteins) is unclear. The authors discuss some interesting potential implications of the new findings (suggested role for CD5 in recruitment of Cbl and Cbl-b to cell membrane, leading to regulation of Lck phosphorylation through CD5-Cbl complex-mediated recruitment of Csk to cell membrane), but there are no experimental data to support these hypotheses. This is a major limitation that severely reduces the significance of the paper.

This point was raised by several reviewers, and we carefully addressed it on pages 1 and 2 of the present point-by-point responses.

Minor points:

1) The discussion is very interesting, but some parts read like a catalogue of interacting proteins. More focus on the novel/critical binding partners could make it easier for the reader.

We tried to prune a bit some of the parts that ‘read like a catalogue’.

2) The authors discuss the possible implications of the finding that Cbl-b, but not Cbl, interacts with LAT. However, a previous paper from the Malissen laboratory (Roncagalli et al. (2014) Quantitative proteomic analysis of signalosome dynamics in primary T cells identifies the CD6 surface receptor as a Lat-independent TCR signaling hub), using a similar strategy to characterize the LAT interactome, reported both Cbl and Cbl-b interactions with LAT. Is this difference the result of the current use of more stringent data analysis criteria?

This very issue has been raised by several reviewers and we already addressed it on pages 4 and 5 of the present point-by-point responses.

3) On page 17:

However, the lack of phenotype of PAG-deficient mouse together with recent data suggest that additional proteins can compensate for the loss of PAG and contribute to recruit CSK at the inner face of the plasma membrane following TCR stimulation (Dobenecker et al, 2005; Reginald et al, 2015; Xu et al, 2005). Our AP-MS analysis identifies CSK as one of those candidate proteins.

Shouldn't the last sentence be: Our AP-MS analysis identifies CD5 as one of those candidate proteins?

We thank Reviewer #2 for detecting this error. The sentence was corrected in the main revised text.

4) What is the time of TCR stimulation in Figure 7?

The time of stimulation has been added in the revised manuscript (Fig 7 legend).

Reviewer #3:

Comments to the authors:

The CNRS lab from Romain Roncagalli and Bernhard Malissen is at the forefront of TCR signaling network analyses in a systems biology approach. They now have been generated GM mice to express a tagged form of CBL or CBLB, respectively that allowed precipitation by Strep-Tactin-coated beads to identify CBL/CBLB interacting protein candidates as well as determining stimulation-induced dynamic changes in these signalling complexes over time under defined stimulation strength.

Although the basic regulatory steps have been elucidated before, many features of the CBL/CBLB pathway(s) is/are only begin to emerge. Description of the kinetic behaviour and the novel CD5 scaffold function of these E3 ligases thus have made the next step in resolving this rather complex story.

This study is technically well performed and the results reported do fully support the main conclusions. The data are novel and will be of interest for immunologists working on molecular mechanisms of T cell activation.

Taken together, their work represent a very careful study and i.e. based on their in part distinct PPIs, conveys a potentially important message for a subset selectivity of CBL/CBLB pathway(s) functions.

No revision needed.

Reviewer #4:

Summary:

In the paper by Voisinne et al., the authors create mice genetically engineered to express the one-STrEP epitope tag on endogenous Cbl or Cblb. They demonstrate that the T cells from these mice express the epitope tagged Cbl or Cblb at the same levels as Cbl or Cblb in wild type mice and they show that the number and function of T cells is normal. They then use peripheral T cells from these mice to pull down either Cbl or Cblb to assess the interacting partners with these proteins over a time course of T cell receptor activation. The work builds an interacting network of proteins that includes both unique and overlapping proteins within the Cbl and Cblb interaction networks. Their data also have identified numerous interacting proteins not previously described in the BioGrid database. They validate the interactions of CSK, CRKL, and CD5 and Cblb by co-immunoprecipitation experiments. They validate the interactions between CD5 and Cbl by co-immunoprecipitation experiments. They show that upon TCR activation, Cblb co-immunoprecipitates more ubiquitinated proteins than Cbl but interestingly, that when Cbl is precipitated from Cblb null mice, more ubiquitinated proteins co-precipitate with Cbl. Finally they demonstrate that there is less ubiquitinated proteins associated with Cbl or Cblb when they are precipitated from mice that are CD5 null. They conclude that CD5 regulates TCR stimulated Cbl and Cblb mediated ubiquitination.

General remarks:

Cbl and Cblb have been shown to play roles as positive and negative regulators of many signaling systems such as receptor tyrosine kinases and the T cell receptor. Previous work has suggested that the two proteins have differential effects on TCR signaling, in particular Cblb is a potent negative regulator of the costimulatory pathways while Cbl is not. Overall the paper provides a catalogue interacting proteins that associate with Cbl and Cblb upon TCR activation (the majority of the interactions described were increased upon activation) and provide network analyses that allow the reader to see overlap, similarities, and differences in the spectrum of interacting proteins between Cbl and Cblb. There are many previously undescribed interactions and the direct comparison of Cbl and Cblb interactors adds value to this work. This certainly be of interest to investigators studying T cell receptor activation and function and to a lesser extent it will be of interest to investigators studying other systems where Cbl proteins play a role.

Major points:

1. The title of the paper focuses on the role of CD5 as a key regulator of Cbl mediated ubiquitination. CD5 has previously been shown to negatively regulate T cell receptor signaling and further recent papers have shown that Cbl ubiquitinates and down regulates CD5 (Demydenko, BBRC, 392: 400-5, 2010) and that Cbl binds to the cytoplasmic tail of CD5 and that this binding was necessary for the down regulation of VAV, one of the Cbl substrates that is down regulated upon TCR activation (e.g. Roa et al, BBRC, 432:52, 2013). Thus the work showing that both Cbl and Cblb associate with CD5 and that the amount of ubiquitinated proteins that coprecipitate with either Cbl protein is decreased in the absence of CD5 is of interest. However, the work is well short of demonstrating the basis for the increase in the co-precipitation of ubiquitinated proteins with Cbl in the absence of Cblb and elucidating the functional role of CD5 in the observed changes in ubiquitination. Overall the authors need to address more rigorously the changes in ubiquitination that they describe. For example, is the increase in the co-precipitated ubiquitinated proteins seen with Cbl when Cblb is absent due to competition between Cbl and Cblb so that more Cbl associates with CD5 when Cblb is absent (and thus that Cblb is the primary protein responsible for CD5 ubiquitination in wt cells)? The data in Figure 6C suggests more association between Cbl and CD5 when Cblb is absent. What happens to TCR and CD5 ubiquitination and downregulation and signaling when either Cblb or Cbl is lost. Also what happens to TCR ubiquitination and downregulation and signaling when CD5 is lost?

We thank Reviewer #4 for his comment. Although the questions raised above are of great general interest, we believe that these points are out of the scope of our study (see Editor comments on page 2). Indeed, we do not mention or argue that the TCR or CD5 receptors are subjected to CBL-CBLB mediated ubiquitylation. However, as mentioned in the discussion section, we did not observe any CD3 chains in the interactome of CBL or CBLB. Along that line, it should be noted that the impact of CBL deficiency on TCR expression has been only observed in double positive thymocytes. To our knowledge, a direct role for CBL (or CBLB) in the ubiquitylation of the TCR in mature and peripheral T cells (as used in this study) has never been reported.

Regarding CD5, our MS analysis of the CBL and CBLB interactomes did not reveal the presence of any ubiquitylated CD5 peptide. Moreover, we would like to point that, in the present study, we did not observe a degradation of CD5 after TCR stimulation in mature wild-type CD4⁺ T cells (see for instance Figure 1A).

2. The loss of Cbl has minimal effects on TCR signaling in peripheral T cells but Cblb has been implicated to regulate the costimulatory pathway in T cells upon TCR activation. What happens to these pathways when they activate the cells in the absence of CD5?

We thank Reviewer #4 for his/her comment. Concerning the effect of CD5 on TCR signaling in T cells, we would like to refer Reviewer #4 to the response provided to the Editor on pages 2, 3 and 4 where we evaluated the functional role of the novel interaction between CD5 and CSK reported in our study on the antigen-recognition and triggering module of the TCR. Analysis of the impact of CD5, CBL and CBLB on costimulatory signaling pathways was outside the scope of the present study and would require extensive experiments. As hinted in our Discussion, the analysis of the *in vivo* consequences of constitutive CD5, CBL, and CBLB gene inactivation on T cell activation via the TCR and costimulatory molecules has clear limitations since several mechanisms might be set in motion and capable of compensating the missing gene product. Conditional deletion of those genes in mature CD4⁺ T cells will thus be required to appropriately assess those issues.

Minor points:

1. A significant body of data has demonstrated binding of Cbl to LAT and regulation of LAT signaling clusters by Cbl, including downregulation of the TCR and LAT by Cbl. However, the mass spec analysis did not identify LAT in the Cbl interacting networks and the authors

make the point that they found Cblb but not Cbl. This needs to be discussed in . Is it due to the use of peripheral T cells vs thymocytes or is a technical limitation of the study?

This point has been raised by several reviewers and already addressed on pages 4 and 5.

2. The authors do reference the Roa paper cited above as evidence that Cbl associates with CD5 but they do not discuss the fact that the work in that paper also implicates the interaction in down regulation of VAV upon TCR activation. They should include this in the discussion.

We thank Reviewer #4 for his/her remark. Considering that (1) we did not detect VAV in the CBL interactome, (2) that the experiments of Roa were performed in transformed T cells and (3) that they do not allow to conclude on the relative contribution of CBL to VAV down regulation we skipped discussing this issue.

3. Page 11, line 22: in the phrase "Interestingly, in some species, ubiquitin remains fused to S27A following its incorporation in the mature ribosome", would be clearer as "Interestingly, in some species, ubiquitin remains fused to S27A that is incorporated into the mature ribosome.

We thank Reviewer#4 for his/her comment. The sentence was changed in the main text.

4. Page 12, line 7, : the authors should use "large number of..." instead of "large numbers of..."

We agree with Reviewer#4 and have changed the sentence in the main text.

5. Page 13, line 17: "To further refined" should be replaced with "To further refine".

We agree with Reviewer#4 and have changed the sentence in the main text

6. Page 21, line 7: in the sentence "Our AP-MS analysis identifies CSK as one of those candidate proteins", the authors probably meant CD5 instead of CSK.

We agree with Reviewer#4 and have changed the sentence in the main text.

7. Page 49: the legend for Figure 6C has to be switched with the legend for Figure 6D.

We agree with Reviewer#4. Legend of Figure 6 has been corrected.

2nd Editorial Decision

08 June 2016

Thank you again for submitting your work to Molecular Systems Biology. We have now finally heard back from referee #2 who accepted to evaluate the revised study. As you will see the reviewer is now fully supportive and we are satisfied with the modification made. We are pleased to inform you that we will be able to accept your paper for publication in Molecular Systems Biology the following minor amendments:

...- please include Figure R2 into the main manuscript as an Expanded View Figure.

Reviewer #2:

The authors have addressed all the concerns raised by us and by the other reviewers. Most importantly, they have validated several of the predicted PPIs using co-immunoprecipitation, and demonstrated that CD5 expression levels influence Lck Y505 phosphorylation.

One important point is that the new data included in the Figure R2 in the rebuttal are potentially very important. These should be included in the final version of the paper.

2nd Revision - authors' response

14 June 2016

It is a great pleasure to learn about the acceptance of our manuscript for publication in Molecular Systems Biology. As you will see we have performed all the following minor amendments:

...including Figure R2 into the main manuscript as an Expanded View Figure and referring to it in the text and legend of Expanded View Figures.

Corresponding Author Name: Bernard Malissen and Romain Roncagalli

Journal Submitted to: Molecular System Biology

Manuscript Number: MSB-16-6837